# Age-dependent predictors of effective reinforcement motor learning across childhood

Nayo M Hill[1,2]*, Haley M Tripp[2], Daniel M Wolpert[3,4,5]†, Laura A Malone[2,6,7]†, Amy J Bastian[1,2,6,7]*†

[1]Department of Neuroscience, Johns Hopkins School of Medicine, Baltimore, United States; [2]Kennedy Krieger Institute, Baltimore, United States; [3]Mortimer B. Zuckerman Mind Brain Behavior Institute, Columbia University, New York, United States; [4]Department of Neuroscience, Columbia University, New York, United States; [5]Kavli Institute for Brain Science, Columbia University, New York, United States; [6]Department of Neurology, Johns Hopkins School of Medicine, Baltimore, United States; [7]Department of Physical Medicine and Rehabilitation, Johns Hopkins School of Medicine, Baltimore, United States

*For correspondence: hilln@kennedykrieger.org (NMH); bastian@kennedykrieger.org (AJB)

†These authors contributed equally to this work

Competing interest: The authors declare that no competing interests exist.

## eLife Assessment

This **important** study tests the development of motor reinforcement learning from toddlerhood to adulthood, using a large online sample. They show that learning improves with age in a task that, like real-life movement, involves a continuous range of response options and probabilistic rewards, and link this shift to reduced movement variability and more efficient feedback-based learning through behavioural modeling. Simplifying the task with discrete actions and deterministic outcomes boosted younger children's performance, suggesting early learning is limited by spatial and probabilistic processing. The evidence is **convincing**, although future work may investigate more naturalistic movement.

**Abstract** Across development, children must learn motor skills such as drawing with a crayon. Reinforcement learning, driven by success and failure, is fundamental to such sensorimotor learning. It typically requires a child to explore movement options along a continuum (grip location on a crayon) and learn from probabilistic rewards (whether the crayon draws or breaks). We studied the development of reinforcement motor learning using online motor tasks to engage children aged 3–17 years and adults (cross-sectional sample, N=385). Participants moved a cartoon penguin across a scene and were rewarded (animated cartoon clip) based on their final movement position. Learning followed a clear developmental trajectory when participants could choose to move anywhere along a continuum and the reward probability depended on the final movement position. Learning was incomplete or absent in 3–8 year-olds and gradually improved to adult-like levels by adolescence. A reinforcement learning model fit to each participant identified two age-dependent factors underlying improvement across development: an increasing amount of exploration after a failed movement and a decreasing level of motor noise. We predicted, and confirmed, that switching to discrete targets and deterministic reward would improve 3–8 year-olds' learning to adult-like levels by increasing exploration after failed movements. Overall, we show a robust developmental trajectory of reinforcement motor learning abilities under ecologically relevant conditions, that is, continuous movement options mapped to probabilistic reward. This learning may be limited

by immature spatial processing and probabilistic reasoning abilities in young children and can be rescued by reducing task demands.

## Introduction

In the game of Poohsticks, invented by A. A. Milne and described in *The House at Pooh Corner*, two children each drop a stick into a stream from the upstream side of a bridge (*Milne and Shepard, 1928*). They then race to the downstream side to see whose stick appears first, with the winner scoring a point. The game is repeated with each child trying to find the sweet spot to drop their stick in to win. Given the capricious nature of streams with their turbulent flow, dropping both sticks in exactly the same spot on two successive games can lead to different outcomes. To be an expert, a child must use probabilistic success and failure feedback to select a location along a continuous and infinite set of possible options (anywhere along the span of the bridge) to drop their stick to maximize reward. Despite the complexity of this probabilistic reinforcement task, the current world champion is 9 years old. Here, we examine how children develop the ability to learn such tasks from reinforcement feedback alone.

Reinforcement motor learning is essential for many sensorimotor tasks and is typically driven by (scalar) reward feedback in contrast to vector errors used in adaptation learning. Reward feedback can be simple binary feedback, such as success or failure in hitting the space bar on a keyboard, or continuous, such as the height achieved on a swing. Critically, the learner is not told what to do but must discover which movements produce reward by trying different options (*Sutton and Barto, 2018*). Therefore, a key component of reinforcement learning is exploring and evaluating feedback to maximize reward.

The basic capacity for reinforcement motor learning emerges early in life. For example, a 9-week-old infant will increase kicking frequency when a ribbon connects their foot to an overhead mobile that moves with their kicks (*Rovee and Rovee, 1969*). Three-month-olds can even learn to produce a specific kick height to move a mobile (*Sargent et al., 2014*). Both tasks have a deterministic relationship between the action and the outcome; a correct kick is always rewarded. In a more complex probabilistic reaching task, children aged 3 to 9 years old showed different sensitivities to reward probability (*Stevenson and Weir, 1959*). The youngest aged children were more likely to stick on a rewarded target even if the reward rate was low (e.g. 33%), whereas older children explored other targets. This suggests that younger children are less likely to explore new options and are more willing to accept lower reward rates.

Reinforcement learning has also been studied in cognitive tasks that require children to select from discrete options to receive reward. In a relatively simple task, 2-year-old children could accurately select a reinforced visual image from two choices (*de Sousa et al., 2015*). More complex tasks have been studied in older children and adolescents identifying age-related changes in learning ability (*Xia et al., 2021*; *Cohen et al., 2020*; *Schulz et al., 2016*; *Mayor-Dubois et al., 2016*; *Master et al., 2020*). In one task, participants selected from options in a probabilistic reward environment with hidden factors that could change the outcome. Although children (7–12 years old) and adolescents (13–17 years old) could identify the factors in the environment that changed the outcome, children were unable to use this information to optimize their choices. Similarly, *Schulz et al., 2016* used a sequential learning task (8–25 year olds) to show that probabilistic reasoning improves with age. As a whole, this previous work identifies differences between younger and older children on choice-based selection tasks that require integration of reward feedback to learn successfully.

Sensorimotor tasks involve different combinations of motor and cognitive mechanisms (*Chen et al., 2017*; *McDougle et al., 2016*), and there are both implicit and explicit (cognitive) contributions to reinforcement motor learning (*Holland et al., 2018*; *van Mastrigt et al., 2023*). Cognitive tasks tend to be discrete with arbitrary assignment of reward probability to the explicit choice options (cf. *Giron et al., 2023*). In contrast, motor learning tasks typically have features that are not present in cognitive tasks. Movement choice options can be continuous (akin to the choice of the direction to kick a soccer ball), corrupted by motor noise, and have a spatial gradient of reward (akin to the probability of scoring as a function of kick direction).

Here, we examine motor learning under reinforcement in which we control two key experimental factors. First, rewards can either be deterministic, the same action leads to the same outcome (e.g.

**Table 1.** Participant demographics.

| Task | Kids | | | | Adults | | | |
|---|---|---|---|---|---|---|---|---|
| | CP | DP | CD | DD | CP | DP | CD | DD |
| n | 111 | 106 | 40 | 41 | 33 | 33 | 10 | 11 |
| % Female | 41 | 57 | 38 | 39 | 67 | 58 | 60 | 82 |
| % RH | 95 | 96 | 93 | 78 | 97 | 97 | 100 | 91 |
| **Age (yrs)** | | | | | | | | |
| Mean (std) | 10 (4.2) | 9.5 (4.1) | 5.4 (1.7) | 5.4 (1.5) | 25.2 (4.6) | 24.4 (3.9) | 24.9 (3.7) | 25.9 (4.5) |
| Median | 10 | 10 | 5.5 | 5 | 24 | 24 | 25 | 27 |
| Range | 3–17 | 3–17 | 3–8 | 3–8 | 18–35 | 18–31 | 20–32 | 18–34 |
| **N per age bin** | | | | | | | | |
| 3–5 yrs | 21 | 22 | 20 | 21 | | | | |
| 6–8 yrs | 27 | 21 | 20 | 20 | | | | |
| 9–11 yrs | 20 | 28 | | | | | | |
| 12–14 yrs | 20 | 19 | | | | | | |
| 15–17 yrs | 23 | 16 | | | | | | |
| **Device** | | | | | | | | |
| Mouse | 69 | 61 | 15 | 19 | 17 | 22 | 7 | 8 |
| Trackpad | 16 | 38 | 12 | 14 | 15 | 11 | 3 | 0 |
| Touchscreen | 26 | 7 | 13 | 8 | 1 | 0 | 0 | 3 |

Participants included for each task. *Abbreviations: CP, continuous probabilistic; CD, continuous deterministic; DD, discrete deterministic; DP, discrete probabilistic; n, number; RH, right-handed; std, standard deviation; yrs, years.*

pressing the space bar on a keyboard), or probabilistic, the outcome is stochastic (e.g. the path of a soccer ball depends on the wind and surface on which it is kicked). Second, action options can be discrete (e.g. the space or shift key on a keyboard) or continuous (e.g. the direction of a soccer kick). We report on a series of experiments in which we control both factors —reinforcement feedback (deterministic vs. probabilistic) and the action options (discrete vs. continuous targets) to examine the development of reinforcement learning across childhood. These factors reflect the natural variations in tasks that children have to learn during everyday life and are important factors for rehabilitation interventions. The goal of these tasks is to understand how children at different ages adjust their movements in response to reward feedback. Our study builds on previous work in healthy adults examining center-out reaching movements under binary reward feedback (*Cashaback et al., 2019*; *Therrien et al., 2016*; *Holland et al., 2018*; *van der Kooij and Overvliet, 2016*).

We developed a remote video game paradigm for a cross-sectional study of 298 children aged 3–17 years and 87 adults from across the USA (locations and demographics shown in *Figure 1—figure supplement 1*, *Tables 1 and 2*). We hypothesized that children's reinforcement learning abilities would

**Table 2.** Participant ethnicity and race.

| | Ethnicity | | Race | | | | |
|---|---|---|---|---|---|---|---|
| | Hispanic | Not Hispanic | Black | White | Asian | Multiple | Other |
| Kids n (%) | 19 (6.4) | 279 (93.6) | 15 (5.0) | 234 (78.5) | 17 (5.7) | 24 (8.1) | 8 (2.7) |
| Adults n (%) | 3 (3.5) | 84 (96.6) | 6 (6.9) | 34 (39.1) | 41 (47.1) | 5 (5.8) | 1 (1.2) |

Ethnicity and race classifications were self-reported by participant/parent. Participants who identified as two or more categories of race (Black, White, and/or Asian) were classified as Multiple. Participants who specified Asian (Indian) or South Asian were classified as Asian. Participants who identified as one or more races other than Black, White, or Asian were classified as *Other*.

improve with age and depend on the developmental trajectory of exploration variability, learning rate (how much people adjust their reach after success), and motor noise (here defined as all sources of noise associated with movement, including sensory noise, memory noise, and motor noise). We think that these factors depend on the developmental progression of neural circuits that contribute to reinforcement learning abilities (*Raznahan et al., 2014*; *Nelson et al., 2000*; *Schultz, 1998*). We found that younger children (3–8 years) failed to learn with a continuous target and probabilistic reward feedback. Reinforcement learning improved with age, enabling older children to find the optimal reward region. Using a mechanistic model, we show that this improvement is due to a developmental gradient of increasing exploration after failure and reducing motor noise with age. Importantly, we then show that use of deterministic feedback and discrete targets can dramatically improve learning abilities of younger children.

## Results

Different groups of participants performed one of four tasks: a continuous probabilistic, a discrete probabilistic, a continuous deterministic, and a discrete deterministic task. We first focus on the continuous probabilistic task and its associated reinforcement learning model, as this is the most complex environment, before reporting the results of the other three tasks.

### Continuous probabilistic task

In the continuous probabilistic task, we studied 111 children and adolescents, aged 3–17 years, and 33 adults as they attempted to learn to produce movements that maximized reward. For the key learning phase of the experiment, we created a probabilistic reward landscape in which the probability of reward after each movement depended on its endpoint. To implement this in a task that was engaging to children, we created a computer game that was played at home. Participants were required to move a cartoon penguin from a starting position to join a group of penguins arranged horizontally across the screen (*Figure 1a* - continuous). Participants were told that there was a slippery ice patch just before the group (dark blue area, *Figure 1a* - continuous) and that they should try to make the penguin cross at the location where they would not slip. In the instructions, participants were told that there was a location in the ice where they would never slip. The reward landscape (*Figure 1b* top left) determined the probability of success on each trial and was not known to the participant. There was a small 100% reward zone where the penguin would never slip, with the reward probability decreasing away from this zone. Successful trials led to a short Disney cartoon clip playing, whereas a failed trial led to the penguin falling over and the appearance of a static image of a sad penguin (*Figure 1c*).

Participants performed five blocks of trials (*Figure 2a*). *Figure 2b* shows examples from different aged participants for endpoint position over the experiment. In block 1, participants made 20 movements to a series of discrete targets, which were the same width as the 100% reward zone. The targets appeared at different locations across the scene and participants received reinforcement feedback at the end of each trial (sample paths are shown in *Figure 2—figure supplements 1 and 2*). The targets were presented in a randomized order, but each participant received the same 20 target locations. This baseline block allowed participants to experience reaching to many locations in the workspace and allowed us to assess accuracy and precision in hitting discrete targets. All participants but one performed well in the baseline block of the task, with endpoint position on average within the target zone (*Figure 2c*). This shows that participants could accurately hit discrete targets.

Participants then began the learning block, where they could move to any endpoint location on the continuous target. Note that the reach location for the first learning trial was not affected by (correlated with) the target position on the last baseline trial (p>0.3 for both children and adults, separately). *Figure 2b* shows the 5- and 7-year-old children moved to many endpoint locations (i.e. showed exploration) receiving reward based on the probability landscape. Interestingly, their exploration appeared to show little sensitivity to previous failure (open circles) versus success (filled circles) and endpoints did not converge on the 100% reward zone (gray area) by the end of the block. The older children, aged 11, 12, and 15, performed similarly to the adult, exploring early in the block, and then focusing movements on the unseen 100% reward zone. This indicates that they were able to explore successfully after failures and exploit the 100% reward zone. These patterns were also

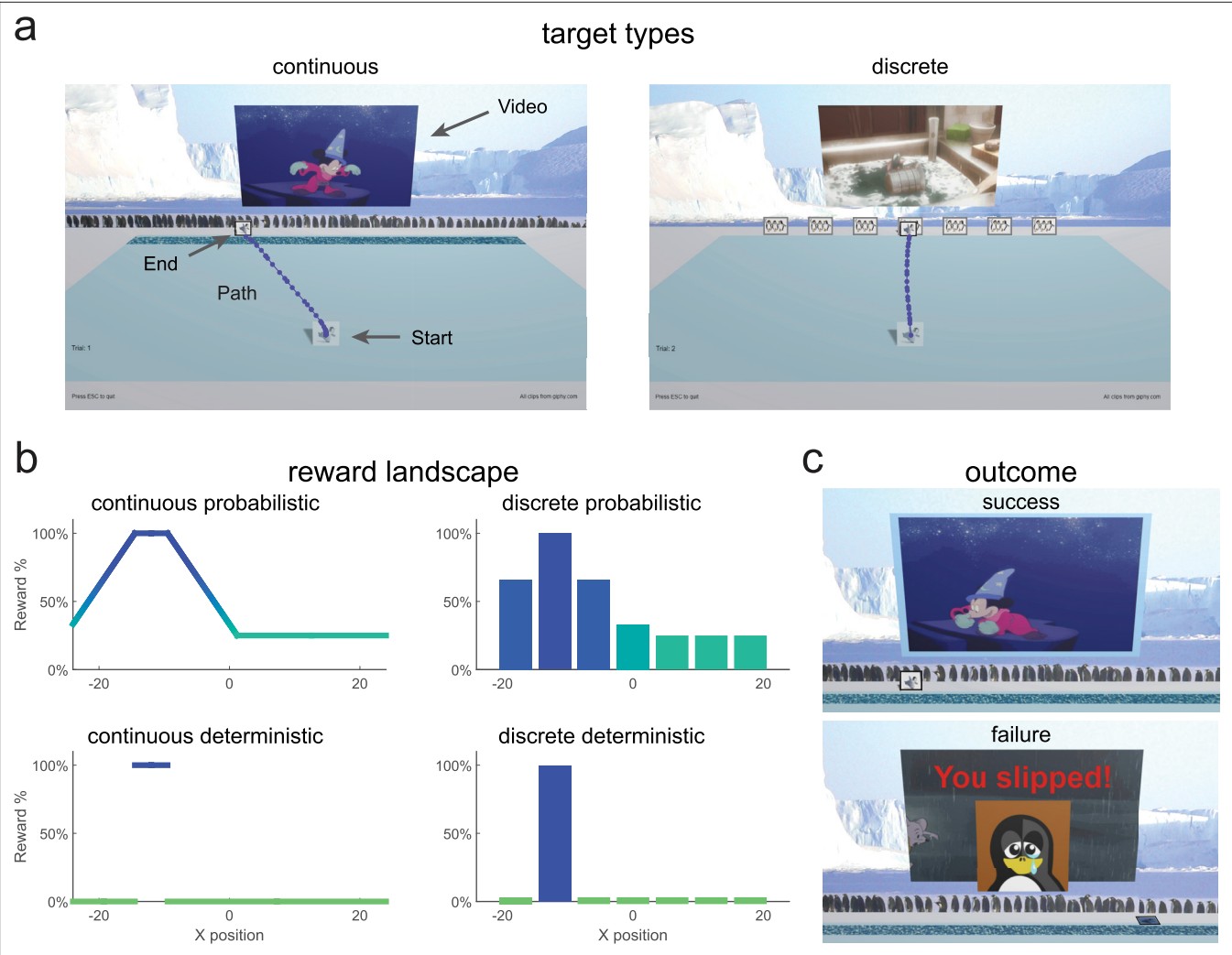

**Figure 1.** Game environment. (**a**) Screenshot of game environment and sample movement path (large text, arrows, and movement path were not displayed to participants). During the learning block, participants either experienced a continuous target (continuous groups) or seven discrete targets (discrete groups). (**b**) Reward landscape for the learning blocks for the different task paradigms. The x-axis represents normalized units based on the participant's computer setup. Continuous probabilistic: continuous target with position-based reward probability gradient; discrete probabilistic: discrete targets with target-specific reward probabilities; continuous deterministic: continuous target with a single 100% rewarded zone; discrete deterministic: discrete targets with a single target giving 100% reward. (**c**) Outcome feedback for continuous probabilistic task. Success (top), movie clip (different for each trial), and pleasant sound plays, and video screen is outlined in blue. Failure (bottom), movie clip does not play, the penguin falls over and red text 'You slipped!' appears with a sad face image.

The online version of this article includes the following figure supplement(s) for figure 1:

**Figure supplement 1.** Map of participant locations.

observed in group data (*Figure 3*, binned across 6 age groups) where, on average, the 3–5 and 6–8 year-olds did not find the 100% reward zone, but the older age groups did.

After learning, participants transitioned seamlessly (no break or change in instructions) into two short blocks with clamped feedback to examine behavior in response to repeated success or repeated failure. The success clamp always rewarded movements, and the fail clamp never rewarded movements. The 5- and 7-year-olds explored similarly in both clamped blocks whereas the older children showed greater exploration in the fail clamp compared to the success clamp (*Figure 2b*).

In the final block, participants moved to a single discrete target in the center of the workspace to examine whether the precision changed over the course of the experiment (examples *Figure 2b*; group data *Figure 2d*). For participants aged 3–17 years, an ANOVA of accuracy (distance from discrete target center) by block (first vs. last) and age (5 bins) shows no effect of block ($F_{1,106}$=0.282,

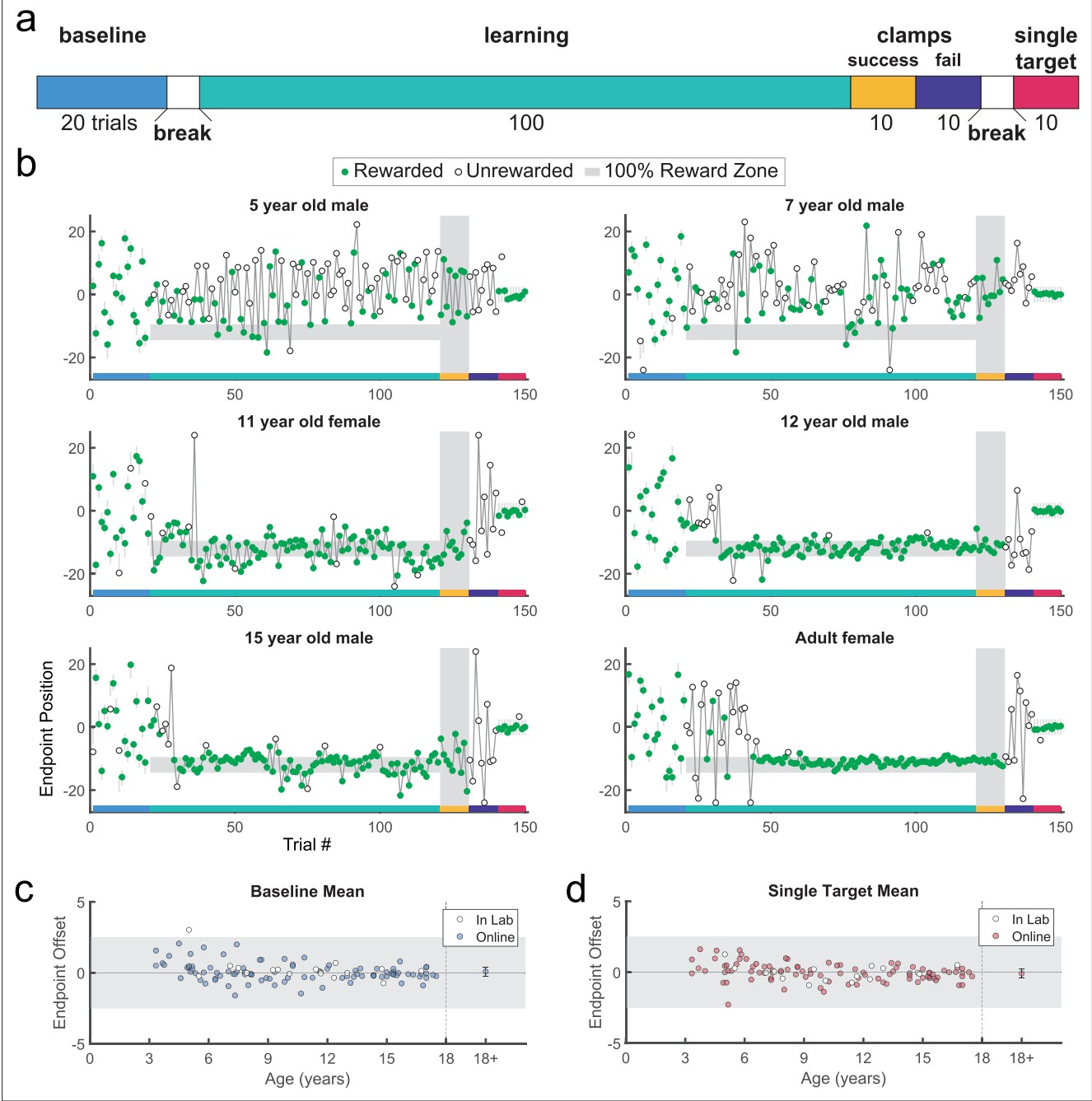

**Figure 2.** Paradigm, example behavior, and target accuracy in the continuous probabilistic task. (**a**) Experimental design with baseline: single discrete target presented in randomized locations across the screen; learning: learning block with reward determined by endpoint position; success clamp: feedback clamped to 100% success independent of endpoint position; fail clamp: feedback clamped to 100% failure independent of endpoint position; single target: single discrete target presented in the middle of the screen. (**b**) Representative endpoint time series from various aged participants. Gray shaded zones indicate positions in the workspace where a reward is given 100% of the time (thin gray lines in first and last blocks are for the discrete targets). Green filled circles indicate rewarded trials while open circles indicate unrewarded trials. The horizontal colored bar on the x-axis indicates the trials corresponding with the experimental blocks outlined in (**a**). In the learning block (trials 21–120), rewards were given based on the continuous probabilistic landscape. (**c**) Mean baseline accuracy (average reach deviation from the discrete targets) by age. Adult data (n=33) are averaged and plotted to the right with standard error of the mean. The gray region shows the width of a discrete target. (**d**) Same as (**c**) for the single target in block 5. In (**c**) and (**d**), participants who completed the task in person (in lab; n=16) are indicated in white circles.

The online version of this article includes the following figure supplement(s) for figure 2:

**Figure supplement 1.** Example baseline paths for participants ages 3–11 years old.

*Figure 2 continued on next page*

*Figure 2 continued*

**Figure supplement 2.** Example baseline paths for participants ages 12–17 years old and adults.

**Figure supplement 3.** Path length ratios.

**Figure supplement 4.** Timing information.

p=0.597) and no interaction between age and block ($F_{4,106}$=1.219, p=0.307), indicating that accuracy of movement to discrete targets was maintained throughout the experiment even for the younger children.

Note that participants were instructed to reach to the continuous target; there were no instructions regarding the path nor the precise timing of the movements. In younger children, movements tended to be more curved compared to older children, as indicated by larger path length ratios (*Figure 2—figure supplement 3*). Movement times were similar for all children (*Figure 2—figure supplement 4*), but younger children took longer to initiate a trial by clicking on the ball (longer Reaction Time) and start moving after initiating (longer Stationary Time). Since these elements of movement were not

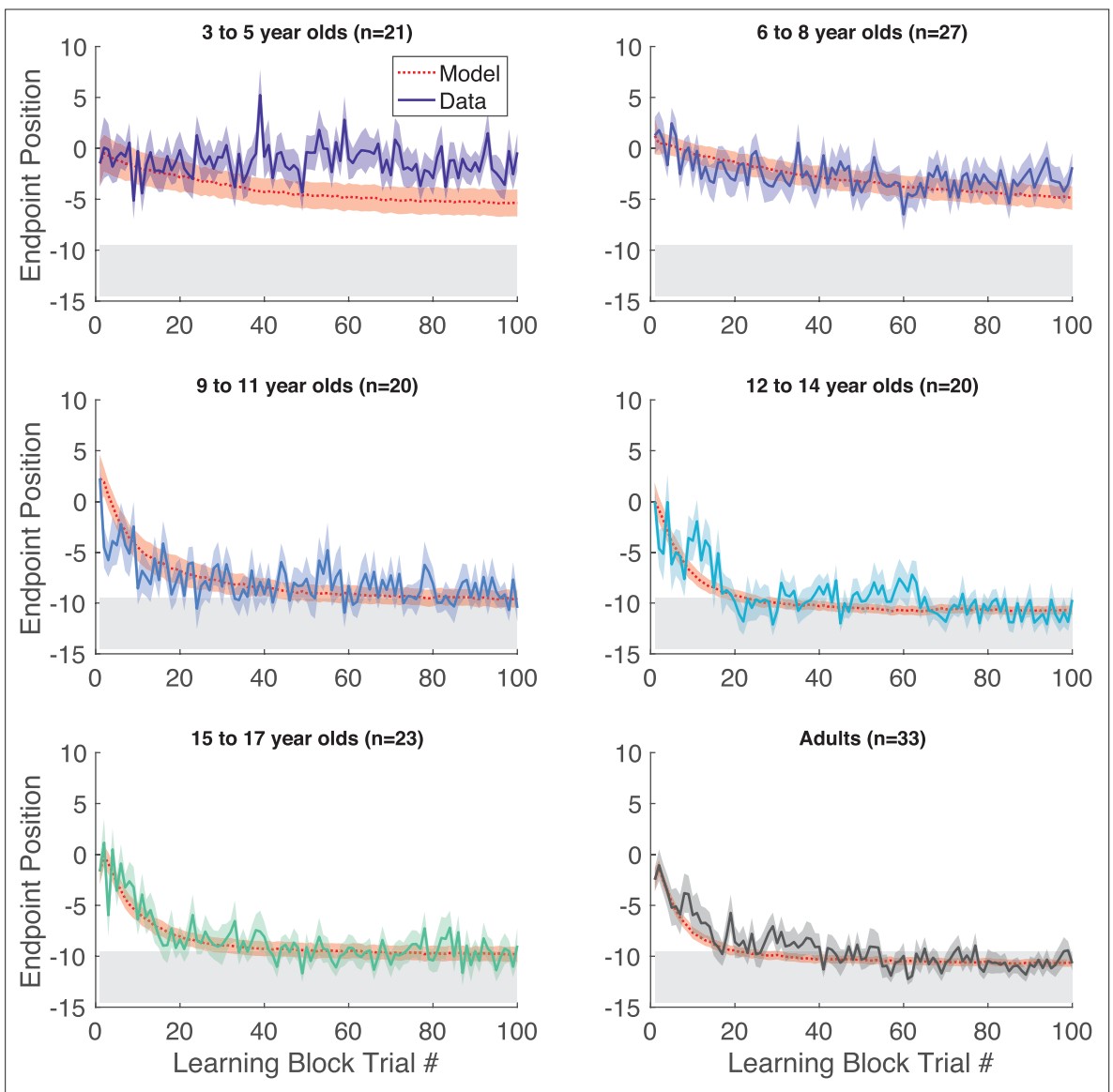

**Figure 3.** Continuous probabilistic task learning block time series. Data and model (red, smooth curves) for each trial of the learning block grouped into age ranges. Data shows mean (solid line) and standard error of mean (shading) of participants' endpoint. The model in red shows mean (dashed lines) and standard error (shading) from the model simulations. The gray region shows 100% reward zone.

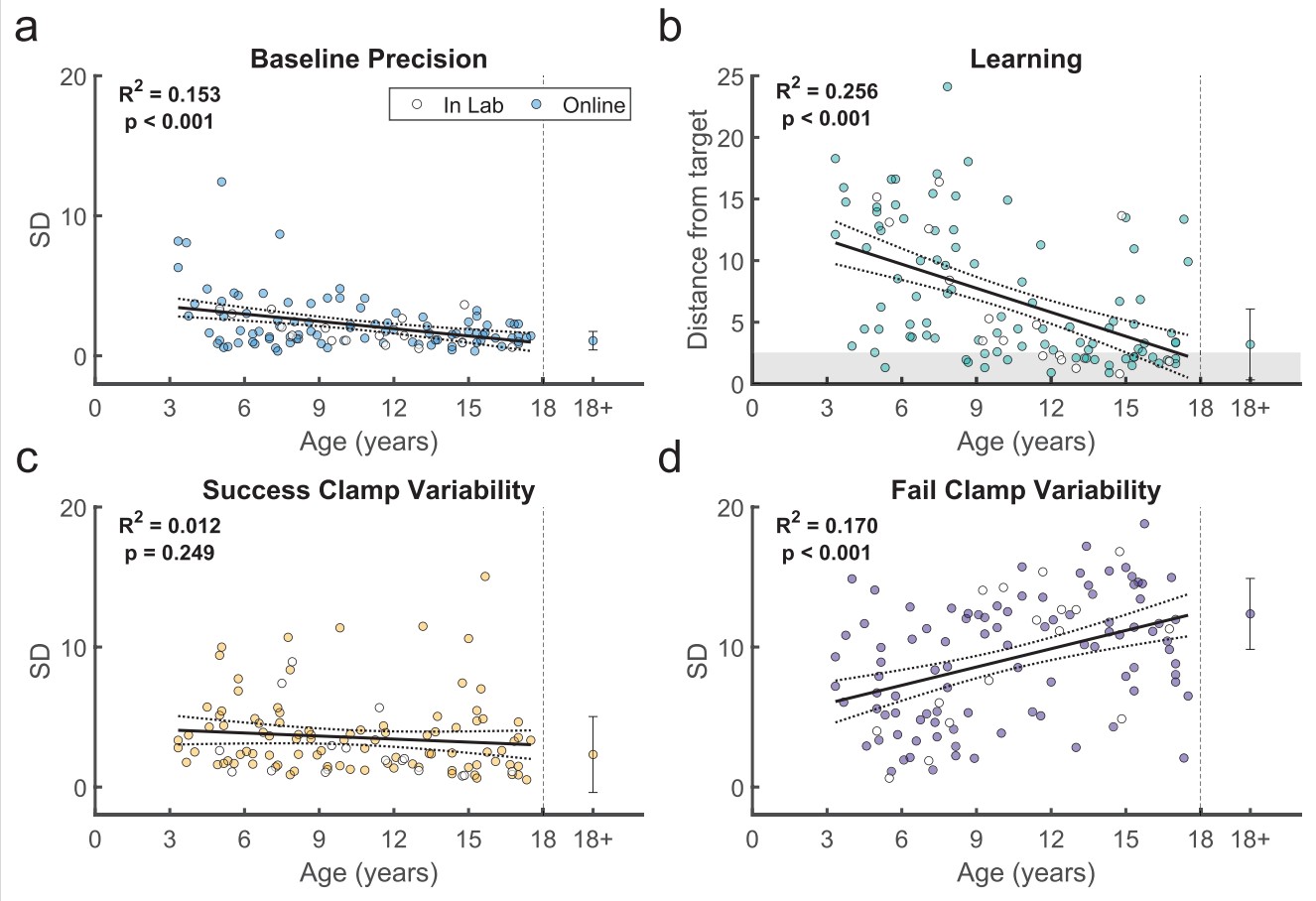

**Figure 4.** Variability and learning in the continuous probabilistic task. (**a**) Baseline precision by age. Average adult variability shown for comparison. (**b**) Learning block performance by age. Learning measure is the distance from the target, measured as the absolute distance from the 100% reward zone. (**c**) Endpoint variability in the success clamp by age. (**d**) Endpoint variability in the failure clamp by age. Regression line with 95% confidence interval shown for children and error bars show standard error of the mean for adults (n=33). Participants who completed the task in person (in lab; n=16) are indicated in white symbols.

instructed and did not prevent participants from hitting discrete targets, we focus on the endpoint of the movement in our analyses.

## Measures associated with learning in the continuous probabilistic task

We assessed how key features of behavior changed across childhood (*Figure 4*, with the average of adult performance for comparison). In the Baseline block, the endpoint variability relative to target center (*Baseline precision*) decreased significantly with age (*Figure 4a*; regression $R^2 = 0.153$, $F_{1,109} = 19.7$, p<0.001). This is consistent with known developmental changes in motor variability (*Takahashi et al., 2003*; *Deutsch and Newell, 2005*) and may represent one component of motor noise that is distinct from exploration.

Children also improved learning as a function of age by reducing their endpoint distance from the 100% reward zone (*distance from target*). The distance from the target, averaged over the last 10 reaches, significantly reduced with age (*Figure 4b*; regression $R^2 = 0.256$, $F_{1,109} = 37.4$, p<0.001). Younger children rarely reached the 100% reward zone (gray region), whereas children over 9 years old often did.

We used the clamp blocks to determine how participants responded to success versus failure, and if this could contribute to developing adult-like learning patterns. We would expect participants to explore more after failure versus success. To assess this, we calculated the standard deviation of movement endpoints in each clamp condition as a measure of movement variability (*Figure 4c & d*). Adults showed high variability after failure and low variability after success, as expected. In children,

variability after failure was low in younger children and increased significantly with age (regression $R^2 = 0.17$, $F_{1,109} = 22.3$, p<0.001), doubling across the age range to reach adult levels (*Figure 4d*). In contrast, variability after success was relatively stable across age (regression $R^2 = 0.012$, $F_{1,109} = 1.34$, p=0.249). Overall, these results suggest that younger children do not explore as much as older children after failures, which is essential to finding the reward zone.

## Comparing reinforcement learning models of the task

To understand how participants update their reach endpoint based on binary feedback, we developed a set of reinforcement learning models of the task. Several reinforcement models have been proposed for tasks similar to ours (*Therrien et al., 2018*; *Therrien et al., 2016*; *Cashaback et al., 2019*; *Roth et al., 2023*). In the full model, a participant maintains an estimate of their desired reach location ($x_t$) which can be updated across trials. On each trial, the actual reach location ($s_t$) is the desired reach with the addition of (possibly) exploration variability ($e_t$), planning variability ($p_t$) and motor noise ($m_t$). Motor noise, as we use it here, can include other sources of noise that do not benefit movement (e.g. sensory and memory noise). The distinction between exploration and planning variability is that exploration is only added if the last trial was unsuccessful (reward $r_t = 1$ for successful trials, and 0 for unsuccessful trials) whereas planning variability is added on all trials:

$$s_t = x_t + (1 - r_{t-1})e_t + p_t + m_t \tag{1}$$

where the sources of variability are all drawn from zero-mean Gaussians with standard deviations given by $\sigma_e$, $\sigma_p$ and $\sigma_m$.

The probability $p$ of receiving a reward, $r_t$, depends on the actual reach endpoint and the particular reward regime used such that the *reward* equation is

$$p(r_t = 1) = f(s_t) \tag{2}$$

where $f()$ can represent different functions such as the continuous or discrete probabilistic and deterministic reward regimes. The desired reach can then be updated by both planning noise and any exploration noise but not by motor noise:

$$x_{t+1} = x_t + \eta_e r_t (1 - r_{t-1})e_t + \eta_p r_t^p p_t \tag{3}$$

where $\eta_p$ and $\eta_e$ are the learning rates controlling exploration and planning contribution to the update. In the full model $r_t^p$ is the reward on that trial $r_t$ so that both planning and exploration are used to update only after success. In some model variants $r_t^p = 1$ so that the desired reach is always updated by planning variability. By allowing different sources of variability, learning rates, and update rules, there are 30 reduced variants of the full model allowing us to fit all 31 models to the data (see Methods and *Figure 5—figure supplement 1*).

To identify the best model variant that could account for the data, we performed model selection. We fit the model to the 100 learning trials for each child (see Methods for details), so as not to contaminate model selection by behavior in the clamp blocks or of the adults. We performed BIC comparison combining the BICs from the continuous and discrete probabilistic tasks (which had the largest datasets). Model 11 was the preferred model with a ΔBIC = 216 to the next best model (*Figure 5—figure supplement 1*). The same model was preferred when we only used BICs from each of the two tasks separately for either the children alone or including the adults. We also considered a value-based model that tries to learn the value of each location. This model is based on the model by *Giron et al., 2023* (See Methods). This model was not preferred to Model 11 (ΔBIC = 3018) when fitting the children in the Continuous probabilistic task.

## Reinforcement learning model with exploration variability and motor noise

In the preferred model (Model 11 which, from now on, we will refer to simply as the model) there was no planning noise and the learning rate for exploration variability was unity. This model is a special case (with no exploration after success) of the model proposed in *Therrien et al., 2018* and has the following output and update processes:

$$s_t = x_t + (1 - r_{t-1}^e)e_t + m_t \tag{4}$$

$$x_{t+1} = x_t + r_t e_t \qquad (5)$$

In this model, if the previous trial was a failure (*Figure 5a*, top), the current reach is the desired reach with the addition of exploration variability and motor noise. The desired reach for the next trial is updated only if the current reach is successful. In contrast, if the previous reach was a success (*Figure 5a*, bottom), the participant does not explore so that the current reach is the desired reach

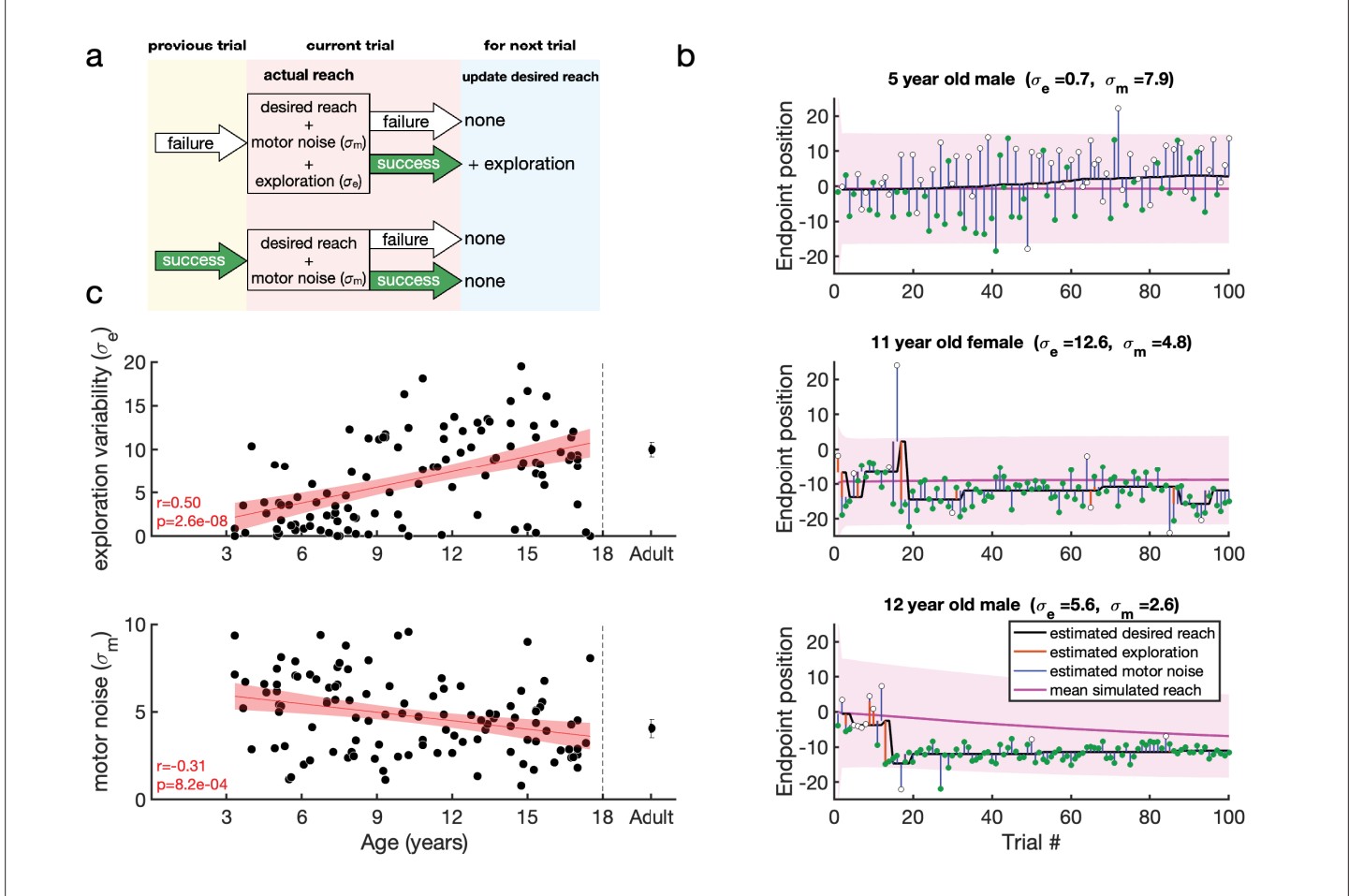

**Figure 5.** Reinforcement learning model for the continuous probabilistic task. (**a**) Model schematic. The participant maintains an estimate of the desired reach which they can update across the experiment. The actual reach on the current trial (pink box) depends on whether the previous trial (yellow box) was a failure (top) or success (bottom). After failure (top), the actual reach is the desired reach with the addition of exploration variability and motor noise (draws from zero mean Gaussian distributions with standard deviations $\sigma_e$ and $\sigma_m$, respectively). In contrast, if the previous trial was a success (bottom), the participant does not explore so that the actual reach is the desired reach with only motor noise. The actual reach determines the probability of whether the current trial is a failure or a success. If the current trial is a success, the desired reach is updated for the next trial (blue box) by the exploration (if any). (**b**) Examples of model fits to three participants. The data are shown as circles, with success trials filled green and unsuccessful trials filled white. The estimated desired reach is shown as a thick black line, and the estimated exploration variability (orange line) and motor noise (blue line) connect the desired reach to the data. The simulation of the participant with the fit parameters is shown in the pink line with shading showing one standard deviation across the simulations. (**c**) Model fit parameters $\{\sigma_m, \sigma_e\}$, by age for the continuous probabilistic group. The line is a regression fit (with 95% confidence interval in shading) to the data for participants younger than 18 years old. The correlation and p-value for each regression are shown in the bottom left corner of each plot (and exclude the adult data). Average adult (n=33) parameters are shown on the right with standard error of the mean.

The online version of this article includes the following figure supplement(s) for figure 5:

**Figure supplement 1.** Model comparison for the continuous and discrete probabilistic tasks.

**Figure supplement 2.** Model parameter recovery.

**Figure supplement 3.** Example fits of the model to the continuous probabilistic task.

**Figure supplement 4.** Fits to the success and failure clamp phases for the continuous probabilistic task.

with motor noise. As there is no exploration, the desired reach will remain the same for the next trial whatever the outcome.

To gain more power for parameter estimates, we then used this model to fit the 120 trials in the experiment (learning and clamp phases). Having done this, we then examined how well our model fitting procedure could recover parameters from synthetic data. This analysis showed that both parameters ($\sigma_e, \sigma_m$) were well recovered with correlations with the true parameters of at least 0.97 (*Figure 5—figure supplement 2*).

Example fits of the model are shown in *Figure 5b* for three participants. The first participant (top) is an example of a 5-year-old non-learner with minimal exploration variability and large motor noise. The desired reach (black line) is connected to the data (filled and hollow circles for rewarded and unrewarded trials) by the estimates of motor noise (blue vertical lines). The other participants (bottom) show learning. Here the desired reach is connected to the data by either motor noise after a successful reach, or by both motor noise and exploration variability (orange vertical lines) after an unsuccessful reach. The desired reach only changes after a successful trial that was preceded by a failure trial (hence exploration). The pink lines and shading show the mean feedforward simulation and standard deviation. *Figure 5—figure supplement 3* shows individual fits to the same participants as in *Figure 2*.

We examined how the two parameters fit to the data varied with age (*Figure 5c*). This showed that both parameters varied significantly with age for the children (both p<0.001). Motor noise decreased by about 40% within the age range tested. Exploration variability increased with age, with an almost fourfold increase on average from the youngest to the oldest child. While this change is in the same direction to the variability in the fail clamp, the model allows us to examine exploration variability in the learning setting and separate it from motor noise. Overall, this model analysis suggests that the increased motor noise and reduced exploration limit the ability of the younger children to learn efficiently.

The simulations of the model with the fit parameters accounted for the differences in learning across the age group bins (*Figure 3* red line and shading show model simulations). While the BIC analysis with the other model variants provides a relative goodness of fit, it is not straightforward to provide an absolute goodness of fit for probabilistic models such as the one we use here (as explained in the Methods). To provide an overall goodness of fit, we examined the traditional $R^2$ between the average of all the children's data during learning and the average simulation of the model (repeated 4000 times) for each child so as to reduce the stochastic variation. This analysis gave an $R^2 = 0.41$.

We compared the model fits for the reach variability with the empirical data in the two clamp blocks. *Figure 5—figure supplement 4a and b* show the variability expected from simulations of the model (red) and the measured movement variability of the children (blue) for the success and fail clamps as a function of age. We also examined the model fits against the empirical data (*Figure 5— figure supplement 4c and d*), which showed we could explain 16% and 42% of the variance for the success and fail clamp phases (note that we only have 9 trials to calculate each clamp block standard deviation).

Overall, the results from the continuous probabilistic task model results suggest that through development, children's motor noise decreases, exploration variability increases, and upon success, the desired reach is fully updated by any exploration present.

## Discrete probabilistic task

Thus far, we identified features that limit learning in young children within a continuous probabilistic landscape. In a second experiment, we asked if segmenting the continuous target into seven discrete targets could improve learning (*Figure 1a* – discrete). In this new task, participants were not able to move between targets but were required to definitively reach to a single target on each trial. We predicted that discrete targets could increase exploration by encouraging children to move to a different target after failure. We studied a new cohort of 106 children and 33 adults as they learned the discrete probabilistic task (*Figure 1b* top right).

Individual example data show that the older children tend to move to a new target after failure, whereas the 3-year-old often chose the same target after failure (*Figure 6—figure supplement 1*). Discretizing the target appeared to improve the learning for the two youngest age groups (*Figure 6a*). Learning, baseline precision, and fail clamp variability had similar trends across age as the continuous probabilistic group (*Figure 6—figure supplement 2*). That is, distance from target decreased with

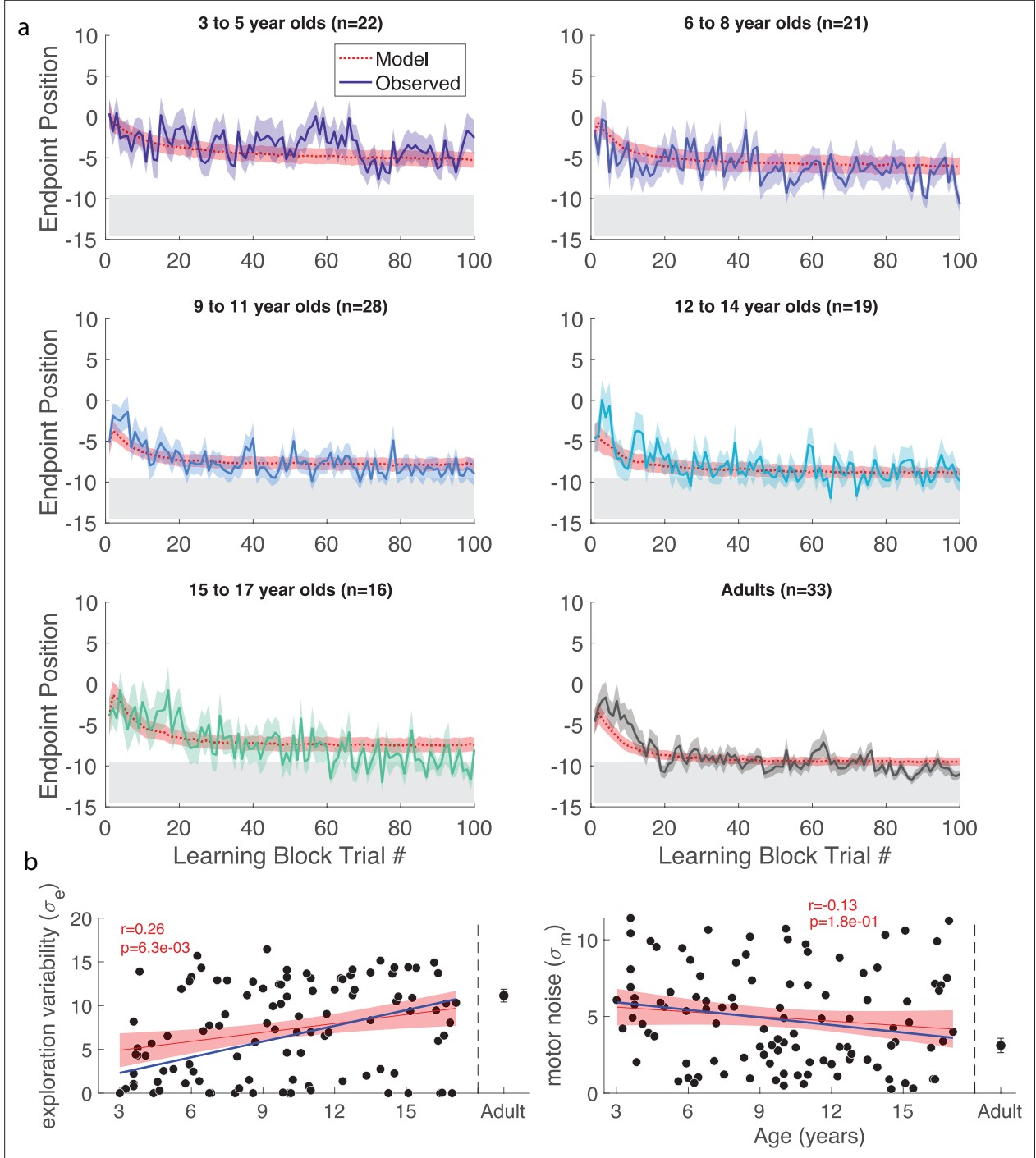

**Figure 6.** Discrete probabilistic task learning block time series and parameter fits. (**a**) Same format as *Figure 3*. (**b**) Panels in same format as *Figure 5c* with the regressions for the continuous probabilistic task overlaid in blue.

The online version of this article includes the following figure supplement(s) for figure 6:

**Figure supplement 1.** Example behavior and discrete target performance in the discrete probabilistic task.

**Figure supplement 2.** Variability and learning in the discrete probabilistic task.

**Figure supplement 3.** Example fits of the model to the discrete probabilistic task.

**Figure supplement 4.** Fits to the success and fail clamp phases for the discrete probabilistic task.

age ($R^2$=0.074, $F_{1,104}$ = 8.31, p=0.0048), baseline precision improved with age ($R^2$=0.150, $F_{1,104}$ = 18.4, p<0.001) and fail clamp variability increased with age ($R^2$=0.162, $F_{1,104}$ = 20.1, p<0.001). In addition, success clamp variability decreased with age ($R^2$=0.048, $F_{1,104}$ = 5.25, p=0.024).

We assessed if there were differences in these measures between the discrete and continuous probabilistic groups by comparing regressions as a function of age, with the intercept set to the age of the youngest child (*Figure 6—figure supplement 2* vs. *Figure 4*). This showed there was no significant difference in baseline precision or variability after success. However, the variability after failure had a higher intercept (increase of 41%; p=0.048) for the discrete probabilistic group, consistent with increasing exploration in the younger children. In addition, the intercept was lower for the distance to target (decrease of 41%; p=0.007), consistent with better learning in the younger children with discrete targets.

## Modeling the discrete probabilistic task

We fit the model to the discrete probabilistic task. Exploration increased significantly with age (p<0.001; *Figure 6b*), but the motor noise change with age was not significant. To evaluate the fit parameters by age between the two tasks, we compared the linear regressions (*Figure 6b* blue line is continuous regression replotted on the discrete parameters). This showed that neither slope nor intercept (set to the youngest child's age) was significantly different between the groups. In sum, the empirical data showed a significant increase in variability after failure for the younger children. The exploration parameter of the model changed in the expected direction but did not reach significance.

## Deterministic reward tasks

In both experiments considered so far, the reward was probabilistic. While this provides gradient information that should guide the participant to the target, younger children may not use the same strategy for this reward landscape. In the final two tasks, we studied new cohorts of 3–8 year-old children since they showed poorest learning in the continuous and discrete probabilistic tasks. We assessed the effect of a deterministic reward landscape (*Figure 1b*, bottom row) on learning with the continuous and discrete targets.

*Figure 7a* compares the time courses for the 3–8 year olds across all four tasks. There was no significant difference in mean age between the four tasks ($F_{3,168}$=1.072, p=0.362) nor was there a significant interaction between task and sex on learning ($F_{3,164}$=0.97, p=0.409). An ANOVA of final learning (*Figure 7* left panel) showed a significant effect of Target type (discrete better than continuous; $F_{1,168}$=12.87, p<0.001) and Reward landscape (deterministic better than probabilistic; $F_{1,168}$=43.66, p<0.001) with no interaction ($F_{1,168}$=0.24, p=0.628). Learning was worst under the continuous probabilistic task, followed by the discrete probabilistic task and continuous deterministic task. The 3–8 year olds performed best on the discrete deterministic task. This shows that making the task discrete or deterministic improves learning and that these factors were additive.

This improvement in learning was not due to significant differences in baseline precision (*Figure 7b* 2nd panel; ANOVA gave Target type: $F_{1,168}$ = 0.16, p=0.69, Reward landscape: $F_{1,168}$ = 0.56, p=0.455, interaction: $F_{1,168}$ = 1.71, p=0.192). For the success clamp block, there was no main effect of Target type ($F_{1,168}$=0.07, p=0.79), but a main effect of Reward landscape ($F_{1,168}$=6.71, p=0.01; less variability with a deterministic landscape) with no interaction ($F_{1,168}$=3.58, p=0.06). This is consistent with there being no advantage to explore after success under a deterministic reward landscape, whereas there is under a probabilistic landscape (exploration can lead to locations that have higher probability of reward). For the fail clamp block, there was a main effect of both Target type ($F_{1,168}$=29.93, p<0.001; greater variability for discrete targets) and Reward landscape ($F_{1,168}$=9.26, p=0.003; greater variability with a deterministic landscape) with no interaction ($F_{1,168}$=1.01, p=0.316). The increased variability for discrete targets is likely because participants must move to a new target to explore, resulting in a larger position change. Increased variability after failure in the deterministic landscape is likely because a failure at one location predicts there will never be success at that location (in contrast to the probabilistic tasks), thereby encouraging exploration. These show that even young children choose their exploration in a rational way based on task features.

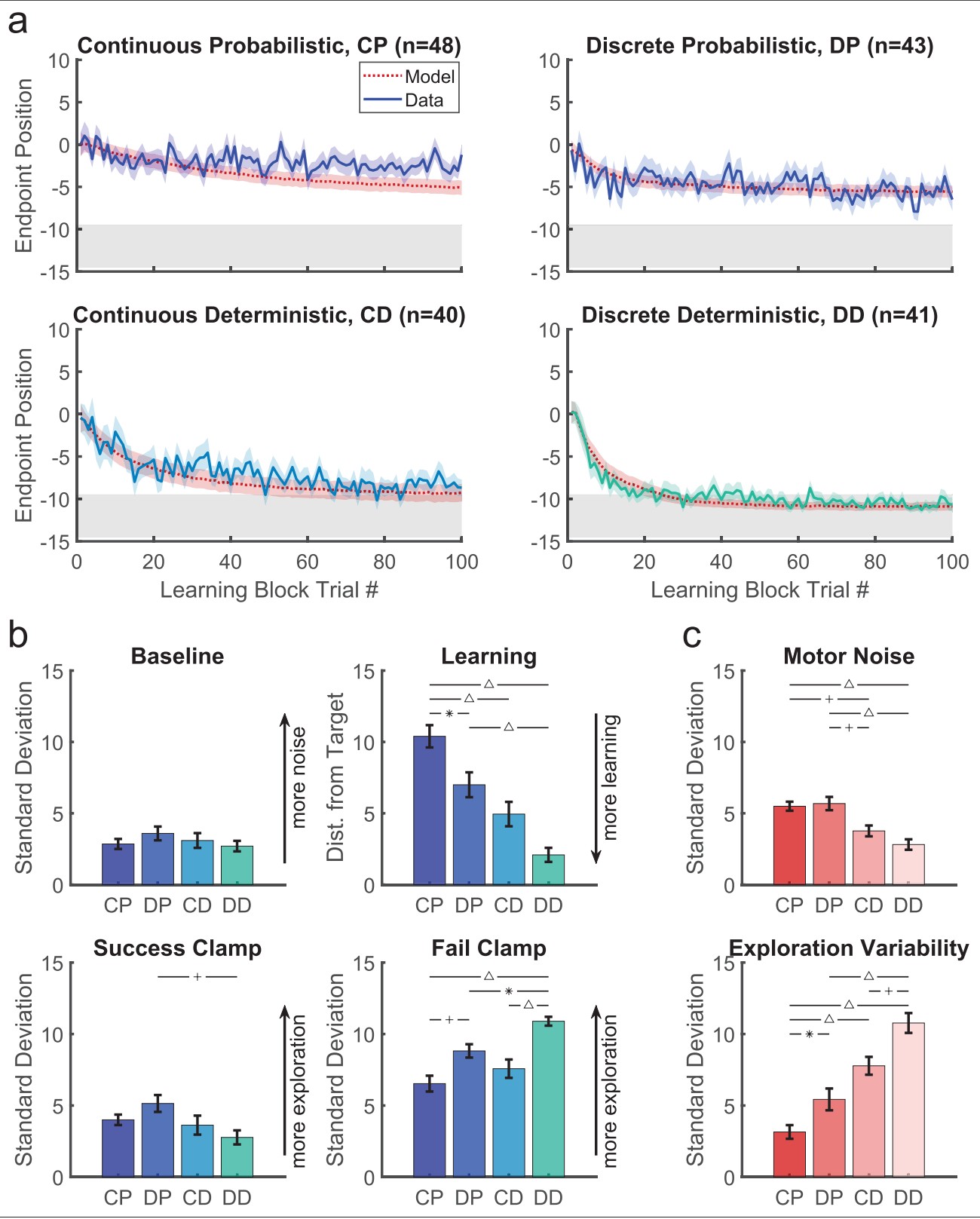

**Figure 7.** Comparison of the four tasks for the 3- to 8-year-old children. (**a**) Learning block performance for the continuous probabilistic, discrete probabilistic, continuous deterministic, and discrete deterministic tasks in the same format as *Figure 3*. (**b**) Comparative performance between tasks for precision in baseline, learning distance from target, and variability in the success and fail clamp. Refer to plot titles in (**a**) for sample size per task. Results from an ANOVA showed that learning significantly improved with discrete targets and deterministic reward feedback. Precision in baseline was not

*Figure 7 continued on next page*

*Figure 7 continued*

statistically different between tasks. (**c**) Estimates of motor noise and exploration variability standard deviation from the model. For both (**b**) and (**c**), bars show mean and standard error of the mean. Statistically significant pairwise comparisons indicated as follows: ✳ = p<0.05, + = p<0.01, and Δ = p<0.001. Abbreviations: CD = Continuous Deterministic; CP = Continuous Probabilistic; DD = Discrete Deterministic; DP = Discrete Probabilistic.

The online version of this article includes the following figure supplement(s) for figure 7:

**Figure supplement 1.** Model fit parameters by age for the deterministic tasks.

**Figure supplement 2.** Expected distance as a function of model parameters.

## Modeling the four tasks and the optimal model

Model fits to the 3–8 year olds are shown in *Figure 7a* (red lines and shading). Examination of the fit parameters for these groups (*Figure 7c*) showed that motor noise was significantly affected by Reward landscape ($F_{1,168}$=35.5, p<0.001) but not by Target type. Exploration variability was significantly affected by Reward landscape ($F_{1,168}$=60.1, p<0.001) and Target type ($F_{1,168}$=16.6, p<0.001).

We can use the model to understand the optimal amount of exploration variability for different amounts of motor noise. To do this, we used the model to estimate the average distance to target across the learning phase for different levels of motor noise and exploration variability for the four tasks (*Figure 7—figure supplement 2* colormap). These simulations showed that as motor noise increases, the optimal amount of exploration variability decreases (white lines; see Appendix 1 for a proof of this relation). Although the same trend is seen in the model fits to the children's behavior (gray bars), in general, the children did not produce the optimal amount of exploration.

## Task comprehension

Poor or absent learning in the youngest age group could arise from these participants not understanding the task. However, we can rule out task comprehension as an explanation for lack of learning, at least for the majority of younger participants. First, 16 participants completed the continuous probabilistic task in the lab, and this sample included five participants in the 3- to 8-year-old age range. The task was administered in the same way as the online version, without additional instructions or interaction with the experimenter during the task. However, after completion, the experimenter asked these children to explain what they did during the task as an informal comprehension check, and it was clear that this group understood the task. These in-lab participants were able to accurately describe the goal of the task. Performance for all participants tested in the lab was not significantly different from online completion, and these data points are indicated in *Figures 2 and 4* with white symbols. Wilcoxon signed-rank tests showed no significant difference between in-lab and online participants in baseline mean (Z = 1.238, p=0.216), baseline precision (Z = –0.626, p=0.532), distance from target (Z = –0.348, p=0.728), success clamp variability (Z = –1.952, p=0.051), fail clamp variability (Z = 0.281, p=0.779), or single target mean (Z = –0.399, p=0.690).

Second, we used the reinforcement learning model to assess whether an individual's performance is consistent with understanding the task. In the case of a child who does not understand the task, we expect that they simply have motor noise on their reach, and crucially, that they would not explore more after failure, nor update their reach after success. Therefore, we used a likelihood ratio test to examine whether the preferred model was significantly better at explaining each participant's data compared to the model variant which had only motor noise (Model 1). Focusing on only the youngest children (age 3–5), this analysis showed that 43, 59, 65, and 86% of children (out of n=21, 22, 20, and 21) for the continuous probabilistic, discrete probabilistic, continuous deterministic, and discrete deterministic conditions, respectively, were better fit with the preferred model, indicating non-zero exploration after failure.

In the 3- to 5-year-old group for the discrete deterministic condition, 18 out of 21 had performance better fit by the preferred model, suggesting this age group understands the basic task of moving in different directions to find a rewarding location. The reduced numbers fit by the preferred model for the other conditions likely reflect differences in the task conditions (continuous and/or probabilistic) rather than a lack of understanding of the goal of the task.

## Discussion

We studied a large, cross-sectional cohort of children and adolescents aged 3–17 years old and adults performing reinforcement-based motor learning tasks. Binary feedback—success or failure—was the only learning signal provided as participants moved to continuous or discrete targets under probabilistic or deterministic reward landscapes. The continuous target gave them unlimited movement choices across the target zone, whereas discrete targets constrained choices. The movement endpoint position was mapped to reward probability, with the probabilistic landscape varying from 25% to a small 100% reward region, and the deterministic landscapes had a single, small 100% reward zone. We found a developmental trajectory of reinforcement learning in the continuous probabilistic task, with older children learning more than younger children. This was paralleled by age-dependent increases in exploration after a failed trial, which is essential for learning and decreases in baseline movement precision. A mechanistic model of the task revealed that in addition to the increase in exploration, there was a reduction in motor noise with age.

In contrast to the children in the continuous probabilistic group, younger children learned better when the task had discrete targets or deterministic reward. These effects appeared to be additive—the 3- to 8-year-old children learned best with discrete targets *and* in a deterministic reward landscape. Thus, the youngest children had the fundamental mechanisms for interpreting binary reward signals to drive reinforcement learning of movement—this is not surprising given that this ability has been shown in infants (*Rovee and Rovee, 1969*). However, children aged 3–8 did not effectively utilize reward signals to learn in situations where they had to respond to probabilistic environments or where there were no spatial cues specifying distinct targets.

Our data suggest that the developmental trajectory we identified was not due to poor motor accuracy or lack of understanding of the task in the younger children. We designed the baseline block to ensure that children could accurately hit individual targets presented in different locations across the screen. The width of these targets was the same as that of the hidden 100% reward zone within the learning block, and children of all ages could hit these targets accurately. The youngest children could also learn similarly to adults in the discrete deterministic task. This shows that children were able to understand the concept of the task and how to be successful.

In reinforcement learning tasks, different sources of movement variability have been proposed, together with a range of models of how these sources are used to update the desired reach for future movements. For example, motor noise has been proposed in several models of control (e.g. *Harris and Wolpert, 1998*; *Todorov and Jordan, 2002*; *Trommershäuser et al., 2003*). Such noise is thought to be unintentional and unavoidable, and studies show that humans tend to plan movements to minimize the bad consequences of motor noise. Other sources of variability are typically considered to be beneficial as they can be used to update reaching behavior. For example, in some models, planning variability (*van Beers, 2009*) is always added to the movement independent of whether previous trials were successful or not. A third possible source of variability is exploration that is distinguished from planning variability in that it is only present after an unsuccessful movement. Several models have been proposed with different combinations of motor noise, planning, and exploration variability (e.g. *Therrien et al., 2016*; *Therrien et al., 2018*; *Roth et al., 2023*). We were able to unify these previous models as special cases of a more general model. This allowed us to compare 31 model variants, and we found that the preferred model was a special case (with no exploration after success) of the model proposed in *Therrien et al., 2018*. This suggests that variability in our task is due to motor noise as well as active exploration after failure. The desired reach is updated by the exploration present on a successful trial which was preceded by a failure trial (the necessary condition for exploration). The model differs from the preferred model in *Roth et al., 2023* which also includes a learning rate for the update so that only part of exploration is incorporated into the next reach.

One major finding from this study is that exploration variability increases with age. Some other studies of development have shown that exploration can decrease with age, indicating that adults explore less compared to children (*Schulz et al., 2019*; *Meder et al., 2021*; *Giron et al., 2023*). We believe the divergence between our work and these previous findings is largely due to the experimental design of our study and the role of motor noise. In the paradigm used initially by *Schulz et al., 2019* and replicated in different age groups by *Meder et al., 2021* and *Giron et al., 2023*, participants push buttons on a two-dimensional grid to reveal continuous-valued rewards that are spatially correlated. Participants are unaware that there is a maximum reward available, and therefore children

may continue to explore to reduce uncertainty if they have difficulty evaluating whether they have reached a maxima. In our task, by contrast, participants are given binary reward and told that there is a region in which reaches will always be rewarded. Motor noise is an additional factor that plays a key role in our reaching task but minimal, if any, role in the discretized grid task. As we show in simulations of our task, as motor noise goes down (as it is known to do through development), the optimal amount of exploration goes up (see *Figure 7—figure supplement 2* and Appendix 1). Therefore, the behavior of our participants is rational in terms of increasing exploration as motor noise decreases.

A key result is that exploration in our task reflects sensitivity to failure. Older children make larger adjustments after failure compared to younger children to find the highly rewarded zone more quickly. *Dhawale et al., 2017* discuss the different contexts in which a participant may explore versus exploit (i.e. stick at the same position). Exploration is beneficial when reward is low as this indicates that the current solution is no longer ideal, and the participant should search for a better solution. *Konrad et al., 2025* have recently shown this behavior in a real-world throwing task where 6- to 12-year-old children increased throwing variability after missed trials and minimized variability after successful trials. This has also been shown in a postural motor control task where participants were more variable after non-rewarded trials compared to rewarded trials (*van Mastrigt et al., 2020*). In general, these studies suggest that the optimal amount of exploration is dependent on the specifics of the task.

We found that turning a continuous target into discrete buttons improved learning performance. In our tasks, a child needs to remember their previous endpoint position and whether it was rewarded to decide where to move at the start of the next trial. We suspect that the continuous target task relies more on spatial working memory, which is not fully developed (*van Asselen et al., 2006*), particularly in those under 10 years of age (*Hitch et al., 1988*; *Pickering, 2001*; *Tsujii et al., 2009*).

Younger children may benefit from clear markers in the environment to differentiate various spatial positions along the target. This is consistent with tasks that have assessed children's preferences for reporting information; they find it easier to select from discrete choices on a Likert scale versus using a continuous visual analog scale (*van Laerhoven et al., 2004*; *Shields et al., 2003*). Finally, it is also possible that older children had the ability to use different working memory encoding mechanisms. It is known that phonological encoding using verbal labels develops later than visual encoding and both can be used to hold information in working memory (e.g. my last movement was near the left edge of the screen) (*Pickering, 2001*; *Hitch et al., 1988*). Future work could explore whether providing an explicit verbal strategy to younger children could improve their ability to explore more effectively with a continuous target, highlighting the interplay of cognitive and motor domains in reinforcement learning.

We also found that deterministic reward feedback improved learning. The deterministic landscape is less ambiguous than the probabilistic landscape by design—participants always fail if they are outside the 100% reward zone. The need to track and evaluate the reward frequency across target zones is eliminated, making the task less complex. Younger children have been reported to take longer to choose a reward maximization strategy in a probabilistic environment where all choices have the possibility of being rewarded. *Plate et al., 2018* showed that when children and adults choose between eight options, they initially probability match (i.e. the frequency of selection closely matches the frequency of reward on the various task options). However, adults switch over to a maximizing strategy (i.e. choosing the highest reward option) more quickly than children (*Plate et al., 2018*). In a deterministic landscape, probability matching would result in the same behavior as reward maximizing and therefore the younger children's behavior would appear nearly the same as adults. Young children's behavior may also stem from a fundamental need to prioritize hypothesis testing and gathering information about the world (*Schulz et al., 2019*; *Liquin and Gopnik, 2022*; *Nussenbaum and Hartley, 2019*), a discovery process that facilitates increased knowledge about the environment's causal structure (*Sobel and Sommerville, 2010*).

Our interpretation is that poorer learning performance in tasks besides discrete deterministic was due to an inability to effectively utilize probabilistic reward signals or find the high reward location without clearly delineated spatial cues. This has significant ramifications as most objects in the world do not have delineated targets on them; we learn which location on an object leads to reward by exploring different options (e.g. the best location to grab a mug or push a heavy door open). The world is also not deterministic, as it is rare that the same action will always give the same result. Movement outcomes are probabilistic due to both environmental variation and motor noise (e.g. the

direction of a soccer ball when kicked on different fields or the location of a thrown dart on a dartboard). Eventually, children must learn how to move successfully to interact with their environments using such probabilistic signals.

The differential ability to incorporate reward signals into changes in behavior across childhood may stem from maturation of the reward centers in the brain. Structures important for these processes, such as the basal ganglia reward centers, dorsal lateral prefrontal cortex, posterior parietal cortex, and the anterior cingulate cortex develop throughout childhood (*Raznahan et al., 2014*; *Nelson et al., 2000*; *Schultz, 1998*). As a main underlying neural structure of reinforcement learning, the pallidum reaches peak volume at 9.5 years for females and 7.7 years for males while it takes the thalamus until 13.8 years for females and 17.4 years for males to reach peak volume (*Raznahan et al., 2014*). Older children have more mature brain circuits and may be better able to take advantage of strategies in probabilistic environments that younger children cannot (*Schulz et al., 2019*; *Liquin and Gopnik, 2022*). For example, older children might know to avoid a location as soon as a failure occurs, even if that location was previously rewarded. Younger children might continue sampling those locations, perhaps due to immature signaling in the brain. Indeed, it has been shown that brain activity in younger children can be similar after positive and negative rewards, whereas in older children and adults, it is more distinct (*Mai et al., 2011*; *Eppinger et al., 2009*; *van Meel et al., 2005*).

Online studies such as ours inevitably have some limitations. Given that this task was conducted remotely, we did not have control of the computer operating quality, internet speed, or testing environment for our participants (see Methods for exclusion criteria). As such, we were not able to control or analyze timing parameters of movement in detail, which can be done more easily in in-person experimentation. However, our key analyses compare across large groups that likely factor out these uncontrolled variables. Our results (comparison of in-lab vs. online and fit of a noise model vs. full reinforcement learning model) also confirm that even the youngest age group understood the task. We also recognize that other processes, such as memory and motivation, could affect performance on these tasks; however, our study was not designed to test these processes directly, and future work would benefit from exploring these other components more explicitly.

Our findings in typical development lay a foundation for better understanding of behavior during childhood and could help inform what function may be lost or impaired after injury. This underscores the need to consider not only disease process for interventions but also age, as there are developmental differences in motor learning capacity in individuals at different ages. Knowing how the sensorimotor system works at different ages can guide decisions on how to intervene or alter an environment and give children the best opportunity to use reinforcement learning mechanisms for rehabilitation outcomes.

## Methods

**Key resources table**

| Reagent type (species) or resource | Designation | Source or reference | Identifiers | Additional information |
|---|---|---|---|---|
| Software, algorithm | MATLAB | MATLAB | R2023a | |
| Software, algorithm | Illustrator | Adobe | Version 11.0 | |

### Participants

Children and adults without neurological impairment or developmental delay were recruited to one of four experiments as outlined in *Tables 1 and 2*. A total of 385 complete datasets were included in the analysis. The continuous probabilistic task was designed first, and the other three tasks, discrete probabilistic, continuous deterministic, and discrete deterministic, were conducted to expand upon the results of the first task and further identify factors contributing to learning by reinforcement across the developmental spectrum. Participants were recruited from the Johns Hopkins University community through an online announcement portal, the greater Baltimore Maryland area through Research Match, and nationwide through the online platform Lookit which merged with Children Helping Science in 2023 (*Sheskin et al., 2020*; *Scott and Schulz, 2017*). Our sample includes participants from 38 states out of the 50 United States of America (*Figure 1—figure supplement 1*). Each participant was screened to rule out neurological impairment, developmental delay, and other developmental disorders that could affect motor and cognitive development. This study was approved by

the Johns Hopkins School of Medicine Institutional Review Board and all participants, or their parent/legal guardian, provided informed consent prior to participation.

## Task platform

All four tasks were completed on a web-based platform built with Javascript as previously reported by *Malone et al., 2023* with modifications to the game environment and feedback type. This platform allowed creativity in the game environment design to be kid-friendly and engaging to young participants to foster sustained attention and increase the likelihood of completing the full task. The task platform allowed remote completion of the experiment by participants on their home computer or tablet. A small subset of participants in the continuous probabilistic task (n=16; 3 to 5yo n=2, 6 to 8yo n=3, 9 to 11yo n=5, 12 to 14yo n=5, 15 to 17yo n=1) completed the task in person in the research laboratory and the remainder of participants completed the task remotely. Participants used a mouse, trackpad, or touchscreen input to control a cartoon penguin game piece and move across the game environment. Movement trajectories were sampled at the polling rate of the selected input device, and data from each trial were uploaded and saved to Google Firebase Realtime Database at the end of each trial. Due to the remote data collection nature of this experiment, we were not able to control data sampling rates. Each trial was manually inspected and sampling rates of input devices had a mean of 37.363 ±4.184 Hz.

## Procedure

The game environment is an immersive icy landscape with penguins. The overall goal of the task was to move a penguin from the starting position on one side of the ice (at the bottom of the computer or tablet screen closest to the participant) to a distance of 24 game units (GU) into the game (at the far edge of the screen away from the participant). If successful, a pleasant sound would play, the video screen above the ice would be outlined in blue, and a Disney video clip (different on each trial; gifs hosted on https://giphy.com) would play. Multiple signals of reward were provided to ensure that participants associated their behavior with the feedback provided and could clearly differentiate between a successful trial and a failure trial. The penguin game piece started as a blue color to indicate that it was not active. To activate the penguin, the participant had to click the left mouse button (mouse or trackpad) or touch and hold the penguin, and it would turn white to indicate that it was movable. Then the participant made a reaching movement to move the penguin across the ice. The trial would end when the Y position of the penguin exceeded 24 GU. To account for variability in input device sampling rates, the final X position was computed as an interpolation between the data points immediately before and after Y=24 GU such that every trial had a calculated X position at Y=24 GU.

Rewards were determined based upon the interpolated final X position of the game piece at the back edge of the ice and a task-specific reward landscape. All tasks included five blocks as outlined in *Figure 2a*. Baseline (20 trials): a single, discrete target (image of three dancing penguins) was presented at a randomized X position at the far edge of the ice. Participants were instructed to move accurately to the target. Learning (100 trials): participants were exposed to one of four learning environments with variations in the target type (continuous vs. discrete, *Figure 1a*) and the reward feedback (probabilistic vs. deterministic, *Figure 1b*). In the continuous conditions, participants could choose to move the penguin to any location on a continuous horizontal target (*Figure 1a* – continuous). In the discrete conditions, participants could move to one of seven targets spread horizontally (*Figure 1a* – discrete). For probabilistic feedback, the reward was determined by an unseen position-based probability gradient with a small 100% reward zone away from which reward probabilities decreased linearly to a baseline (*Figure 1b* – continuous probabilistic and discrete probabilistic). For deterministic feedback, reward was always given within a reward zone but not elsewhere (*Figure 1b* – continuous deterministic and discrete deterministic). Success Clamp (10 trials): every trial in this block was rewarded regardless of the final X position of the game piece. Fail Clamp (10 trials): no trials in this block were rewarded. Single Target (10 trials): the same single, discrete target as in baseline was presented in the center of the far edge of the ice, and participants were cued to move accurately towards it. A break for a new set of instructions was provided between Baseline and Learning as well as Fail Clamp and Single Target. Participants were unaware of the reward criteria transitions between Learning and Success Clamp and Success Clamp and Fail Clamp. The ideal/mature behavior in these tasks was to explore at the beginning of the learning block to find the area of most frequent reward

and then exploit this behavior to maximize reward for the remaining trials of the block. Moreover, if the 100% reward zone has been found successfully, continuing to move to this reinforced position during the success clamp and then exploring again in the fail clamp are indicators of mature sensitivity to reward and failure.

## Continuous probabilistic

In the Learning block for the continuous probabilistic task, participants were presented with a continuous target and probabilistic reward feedback (*Figure 1b* – continuous probabilistic). The probability reward landscape is defined by setting reward percentage probabilities at 5 specific $x$ locations with reward between these locations being linearly interpolated (and constant outside the range). We set X values of $\{-24, -14.5, -9.5, 1.1875\}$ to reward probabilities $\{33, 100, 100, 25\}$.

Participants were warned that some of the ice was starting to melt, which would cause the penguin to slip, and were told that in some places they would slip a lot of the time, in some places they would slip some of the time, and in some places they would never slip. They were instructed to move the penguin across the ice without slipping to get the movie clip to play as often as possible. If they were not successful, the penguin would fall over, and they would see a sad face penguin image before the game reset for the next trial (*Figure 1c*). This task design builds from *Cashaback et al., 2019* where participants were asked to reach to any location on a continuous line, with the probability of being rewarded dependent on the reach end location. In their task, there was a small (hidden) 100% reward zone with reward probability decreasing on either side away from this zone. They found that adult participants could learn from a probabilistic reward landscape and find the rewarding zone (*Cashaback et al., 2019*). We explored a similar task design in participants across development.

## Discrete probabilistic

In the Learning block for the discrete probabilistic task, participants were presented with a set of targets, each associated with a specific probability of reward (*Figure 1b* – discrete probabilistic). We set the center of the seven targets at X values of $\{-18, -12, -6, 0, 6, 12, 18\}$ with target width of 5 and with reward percentage probabilities of the $\{66, 100, 66, 33, 25, 25, 25\}$.

The discrete targets were visually the same as those used in the Baseline and Single Target blocks; however, all seven were presented at the same time in equally spaced positions across the edge of the ice. Participants were instructed to find the group of penguins that made the video clip play all the time by moving their penguin game piece across the ice. They were told to move to one group of penguins on each trial and that some penguins would make the movie clip play some of the time, but there was a group of penguins that played the clip all the time. If they were not successful, they would see a red X on the video screen and the video clip would not play before the game reset for the next trial. To ensure that participants could accurately distinguish the feedback associated with each target, there was a visible space between each target. If the participant moved between targets, the participant would receive a message to try again, and the trial would reset until one target was accurately hit.

## Continuous deterministic

In the Learning block for the continuous deterministic task, participants were presented with a continuous target and deterministic reward feedback (*Figure 1b* – continuous deterministic). Participants were warned that some of the ice was starting to melt, which would cause the penguin to slip, and were told that in some places they would slip all of the time and in some places they would never slip. They were instructed to move the penguin across the ice without slipping to get the movie clip to play as often as possible. If they were not successful, the penguin would fall over, and they would see a sad face penguin image before the game reset for the next trial. This task was completed by a subset of participants aged three to eight years and adults.

## Discrete deterministic

In the Learning block for the discrete deterministic task, participants were presented with a set of seven discrete targets and deterministic reward feedback (*Figure 1b* – discrete deterministic). Participants were instructed to move across the ice to one of the groups of penguins on each trial to get the movie clip to play. They were told that one group of penguins would make the video clip play all of the time. If they were not successful, they would see a red X on the video screen and the video clip

would not play before the game reset for the next trial. As in the discrete probabilistic task, to ensure that participants could accurately distinguish the feedback associated with each target, there was a space between each target. If the participant moved between targets, the trial would reset until one target was accurately hit. This task was completed by a subset of participants aged three to eight years and adults.

## Demo versions of tasks

Shortened versions of each task without data collection are provided at the following links. In these versions, there are 5 trials in baseline, 10 trials in learning, 2 trials in each clamp, and 3 trials of single target. To proceed beyond the information input screen, use an arbitrary six-digit code for the subjectID and participant information to sample the game environment.

Continuous Probabilistic:

https://kidmotorlearning.github.io/PenguinsDemo_Continuous-Probabilistic/

Discrete Probabilistic:

https://kidmotorlearning.github.io/PenguinsDemo_Discrete-Probabilistic/

Continuous Deterministic:

https://kidmotorlearning.github.io/PenguinsDemo_Continuous-Deterministic/

Discrete Deterministic:

https://kidmotorlearning.github.io/PenguinsDemo_Discrete-Deterministic/

## Measurements and analysis

We used several metrics to analyze performance and variability in different blocks and evaluate reinforcement learning over childhood. Baseline performance was defined as the signed distance from the target center averaged over each of the 20 baseline trials. Baseline precision was defined as the standard deviation of the baseline performance. Importantly, the width of these baseline targets was the same as the 100% reward zone in the learning block for the continuous target tasks. This allowed determination of both success and precision with moving to targets the width of the goal. Learning performance (*Distance from target*) is defined as the absolute value of the interpolated X position distance from the center of 100% reward zone (located at X = –12 in all tasks) averaged over the last 10 trials of the learning block. A *Distance from target* value closer to 0 indicates better learning. To quantify variability after success, we used the standard deviation of the interpolated X position in trials 2 through 10 of the success clamp block. To quantify variability after failure, we used the standard deviation of the interpolated X position in trials 2 through 10 of the fail clamp block.

Participant characteristics (sex and game play handedness) and game play parameters (device and browser) were analyzed in one way ANOVAs with dependent variables of *Precision* and *Distance from target* to determine whether these parameters influenced learning ability (*Appendix 2—table 1*). To determine whether there was a differential effect of sex, device, or handedness on learning performance in the four different tasks, additional two-way ANOVAs with dependent variable of *Distance from target* were used. There was not a significant interaction between task and sex (reported in Results), task and device ($F_{3,168}$=0.62, p=0.717), or task and handedness ($F_{3,168}$=1.2, p=0.312). Each trial was divided into three phases to describe the reaction, stationary, and movement times. Reaction time is defined as the time from the appearance of the penguin (start of the trial) to the time the penguin is clicked and activated. Stationary time is defined as the time from penguin click to movement onset (first non-zero position). Movement time is defined as movement onset to end of the trial when the penguin moves across the back edge of the ice. Trial timings for each phase of movement were extracted and averaged for each participant across the whole experiment and then averaged within individual age bins to evaluate across ages. Total game play time was also extracted and averaged by age bins. Path length ratios were calculated as the actual path length from the start to end of the trajectory divided by the ideal straight-line path from the first position in the trajectory to the last position in the trajectory. The path length ratio for all trials was averaged for each participant and then averaged within age bins for comparison between ages.

We used linear regression and one- and two-way ANOVAs to evaluate effects of age and other outcome variables on learning as well as compare performance between tasks. Significance level of 0.05 was used for all statistical tests. All raw data processing and statistical analyses were completed using custom scripts in MATLAB (version R2023a) (*Hill et al., 2025*).

## Reinforcement learning model

Several reinforcement models have been proposed for tasks similar to ours (*Therrien et al., 2018*; *Therrien et al., 2016*; *Roth et al., 2023*). With some reparameterizing, all these models can be considered as special cases of a 'full' model. We fit this full model, as well as all 30 possible variants of the full model. These models can be considered mechanistic models as defined by *Levenstein et al., 2023*.

In these models, a participant maintains an estimate of their desired reach location ($x_t$; which reflects the estimated target location) which can be updated across trials. On each trial, the actual real location ($s_t$) is the desired reach with the addition of (possibly) exploration variability ($e_t$), planning variability ($p_t$) and motor noise ($m_t$). The distinction between exploration and planning variability/motor noise is that exploration variability is only added if the last trial was unsuccessful (reward $r_t = 1$ for successful trials, and 0 for unsuccessful trials), whereas planning variability and motor noise is added on all trials:

$$s_t = x_t + (1 - r_{t-1})e_t + p_t + m_t \tag{6}$$

where the sources of variability are all zero mean Gaussian with standard deviations given by $\sigma_e$, $\sigma_p$, and $\sigma_m$. There can be other sources of noise (sensory, memory, etc.) and we regard them as being included as part of motor noise (since they are never used to update the reach). The probability $p$ of reward received, $r_t$, depends on the reach location and the particular reward regime used such that the *reward* equation is

$$p(r_t = 1) = f(s_t) \tag{7}$$

where $f()$ can represent different functions such as the continuous or discrete probabilistic and deterministic reward regimes. The desired reach can then be updated by both planning variability and any exploration noise but not by motor noise (this is the key distinction between planning variability and motor noise).

$$x_{t+1} = x_t + \eta_e r_t (1 - r_{t-1})e_t + \eta_p r_t^p p_t \tag{8}$$

where $\eta_p$ and $\eta_e$ are the learning rates controlling exploration and planning variability contribution to the update. In the full model $r_t^p$ is the reward on that trial $r_t$ so that both exploration (if present) and planning variability are used to update after success. In some model variants $r_t^p = 1$ so that the desired reach is always updated by planning variability.

The variants of the model determine whether (i) each source of variability ($e_t, p_t, m_t$) are present or not, (ii) the settings of the learning rates ($\eta_e$ and $\eta_p$ independent, identical, $\eta_e$ or $\eta_p$ or both set to unity) and (iii) whether $r_t^p$ equals $r_t$ or 1.

This leads to 80 variants: exploration (2) × planning (2) × motor (2) × learning (5) × (2). However, many of these variants can be excluded, leaving 31 model variants. The exclusions are (i) models with no planning variability and no motor noise (as they show no variability in success clamp for example), (ii) models with no exploration variability but a fit or non-zero $\eta_e$, (iii) models with no planning variability but a fit or non-zero $\eta_p$, (iv) models with no planning variability but $r_t^p$ set to $r_t$.

The model variants are shown in *Figure 5—figure supplement 1* and some of these variants correspond to previously proposed models (after allowing renaming of variables; for example, a model with exploration variability present on every trial would be regarded as planning variability). Models 23, 22, 30, 10, and 6 are models 1–5, respectively, from *Roth et al., 2023*. Model 8 is the model from *Therrien et al., 2016* and Model 28 is the model in *Therrien et al., 2018*. Model 20 is the model from *Cashaback et al., 2019*.

## Model fitting

To fit these stochastic models to each participant's data, we used a Kalman smoother. The full model has 5 parameters: $\theta = \{\sigma_m, \sigma_e, \sigma_p, \eta_e, \eta_p\}$. We can reformulate the equations above into a standard Kalman update and output equations. The update equation is given by

$$
\begin{bmatrix} x_t \\ e_t \\ p_t \\ m_t \end{bmatrix} = \begin{bmatrix} 1 & \eta_e r_{t-1}^e & \eta_p r_{t-1}^p & 0 \\ 0 & 0 & 0 & 0 \\ 0 & 0 & 0 & 0 \\ 0 & 0 & 0 & 0 \end{bmatrix} \begin{bmatrix} x_{t-1} \\ e_{t-1} \\ p_{t-1} \\ m_{t-1} \end{bmatrix} + \begin{bmatrix} 0 & 0 & 0 \\ (1-r_{t-1})\sigma_e & 0 & 0 \\ 0 & \sigma_p & 0 \\ 0 & 0 & \sigma_m \end{bmatrix} \begin{bmatrix} v_t^1 \\ v_t^2 \\ v_t^3 \end{bmatrix} \tag{9}
$$

where $v_t^i \sim \mathcal{N}(0,1)$. The output equation is given by

$$
s_t = \begin{bmatrix} 1 & 1 & 1 & 1 \end{bmatrix} \begin{bmatrix} x_t \\ e_t \\ p_t \\ m_t \end{bmatrix} \tag{10}
$$

We initialized the states as all zeros and the state covariance as

$$
\begin{bmatrix} 20^2 & 0 & 0 & 0 \\ 0 & \sigma_e^2 & 0 & 0 \\ 0 & 0 & \sigma_p^2 & 0 \\ 0 & 0 & 0 & \sigma_m^2 \end{bmatrix} \tag{11}
$$

where the wide prior on the initial reach direction allows the Kalman smoother to fit each participant's initial reach. We used Matlab fminsearchbnd to fit the parameters to maximize the likelihood (*Hill et al., 2025*).

## Model selection

We first fit each model to maximize the likelihood over the 100 learning trials for each child participant in the continuous probabilistic task. This ensured that our model selection was not contaminated by behavior in the clamp block. We used BIC to select the preferred model (the BIC provides a relative measure of goodness of fit accounting for differences in the number of parameters of each model). The same model (model 11) was preferred if we only considered the children or also included the adults. Moreover, the same model was preferred for both the children in the discrete probabilistic task (or again if we included the adults). We therefore focus on Model 11 which is a special case (with no exploration after success) of the model proposed in *Therrien et al., 2018*.

## Preferred model

In the preferred model, the produced reach is the desired reach with the addition of the variability from motor noise and exploration (if any), given by the *output* equation:

$$
s_t = x_t + (1-r_{t-1})e_t + m_t \tag{12}
$$

After each reach, the participant updates their desired reach location only if they were successful. They update the reach to incorporate all of the exploration (if any) so that the *update* equation is

$$
x_{t+1} = x_t + r_t(1-r_{t-1})e_t \tag{13}
$$

We examined parameter recovery for the preferred model by generating simulated data (for each participant using their best fit parameters for the preferred model) and compared the fit values with the true values used to simulate the data. This showed correlations were above 0.96 for both parameters.

Having found the preferred model, we used that to fit the 120 trials for each participant in each of the four conditions so as to constrain the parameter fits using all the data. For the discrete tasks, we

fit the data in the same way as the continuous (so that the model does not know about the discrete nature of the task).

## Model simulations

To simulate a participant, we used the best fit parameter to generate Monte Carlo simulations. We used the estimated posterior of the initial state from the Kalman smoother to initialize both the desired and actual reach by sampling from the appropriate posterior 100,000 times and used these to generate the same number of Monte Carlo simulations. We ensured that the desired and actual reaches were constrained to the workspace by clipping values outside the range. We calculate the median (to be robust) of these runs as well as the standard deviation across runs for plotting.

The model simulations also allowed us to simulate the two clamp phases, and we calculated the average standard deviation of actual reaches in each clamp phase across simulations.

## Goodness of fit

While the BIC analysis with the other model variants provides a relative goodness of fit, it is not straightforward to provide an absolute goodness of fit such as standard $R^2$. There are two problems. First, there is no single model output. Each time the model is simulated with the fit parameters, it produces a different output (due to motor noise, exploration variability, and reward stochasticity). Second, the model is not meant to reproduce the actual motor noise, exploration variability, and reward stochasticity of a trial. For example, the model could fit pure Gaussian motor noise across trials (for a poor learner) by accurately fitting the standard deviation of motor noise but would not be expected to actually match each data point and would therefore have a traditional $R^2$ of 0.

To provide an overall goodness of fit, we have to reduce the noise component, and to do so, we examined the traditional $R^2$ between the average of all the children's data and the average simulation of the model (repeated 1000 times per participant) so as to reduce the stochastic variation.

## Optimal exploration variability as a function of motor noise

We used model simulations over a 50 × 50 grid of exploration variability and motor noise to assess the expected learning distance for the 4 tasks. To do this, for each grid location, we simulated each child participant 4000 times (as above, selecting the initial state stochastically for each child) and calculated the expected learning distance over the 100 learning trials across the children. This showed the optimal exploration variability increasing as a function of decreasing motor noise (as also observed with age in the data). For an intuition and proof of this relation, see Appendix 1.

## Alternative value-based model

We considered a value-based model which tries to learn the value of each location. This model is based on the model by *Giron et al., 2023*. In this model, the participant represents the value of each location on each trial $v_t(x)$. Given the reach locations and rewards received up until trials T, $\{x_1, r_1, x_2, r_2 \cdot x_T, r_T\}$ the value function is fit using a Gaussian process with squared exponential kernel and logistic likelihood to give $v_T(x)$. The kernel has two parameters, a length scale $\sigma_l$ which determines the generalization of learning to other locations and a strength $\sigma_s$ that controls the amount of learning (similar to a learning rate). The value function is passed through the softmax function with parameter $b$ that controls the exploration

$$P_t(x) \propto \exp^{bv(x)} \tag{14}$$

where $P_t(x)$ is normalized so as to be the probability of moving to locations $x$.

To fit the models, three parameters $\{\sigma_l, \sigma_s, b\}$ we discretized $x$ into 500 points and used fmincon to maximize the likelihood of the data. We found these optimized parameters were very sensitive to the initial parameter, so we chose 50 random initializations for each participant and selected the one with the highest likelihood. Although we were able to fit the data, unlike the noise-based model, often the parameters were not easily identifiable. For example, there are many ways a participant could fail to learn, such as a large $\sigma_l$ or small $\sigma_s$. Since this was not the preferred model, we chose not to examine the parameters for the fits.

## Acknowledgements

This work was supported by the following funding sources: grants from the National Institutes of Health T32 HD007414 and R35 NS122266 to AJB. DMW is a consultant to CTRL-Labs Inc, in the Reality Labs Division of Meta. This entity did not support or influence this work. NMH gained skills in open science and reproducibility as a learner in the 2024–2025 cohort of Reproducible Rehabilitation funded by NIH NICDH/NCMRR R25 HD105583.

## Additional information

### Funding

| Funder | Grant reference number | Author |
|---|---|---|
| National Institutes of Health | T32 HD007414 | Amy J Bastian |
| National Institutes of Health | R35 NS122266 | Amy J Bastian |
| NIH NICDH/NCMRR | R25 HD105583 | Nayo M Hill |

The funders had no role in study design, data collection and interpretation, or the decision to submit the work for publication.

### Author contributions

Nayo M Hill, Conceptualization, Resources, Data curation, Software, Formal analysis, Investigation, Visualization, Methodology, Writing – original draft, Project administration, Writing – review and editing; Haley M Tripp, Data curation, Visualization, Writing – review and editing; Daniel M Wolpert, Conceptualization, Software, Formal analysis, Supervision, Visualization, Writing – original draft, Writing – review and editing; Laura A Malone, Conceptualization, Supervision, Writing – review and editing; Amy J Bastian, Conceptualization, Resources, Supervision, Funding acquisition, Methodology, Writing – original draft, Writing – review and editing

### Author ORCIDs

Nayo M Hill ⓘ https://orcid.org/0000-0001-9710-0291
Haley M Tripp ⓘ https://orcid.org/0000-0001-9697-1546
Daniel M Wolpert ⓘ https://orcid.org/0000-0003-2011-2790
Laura A Malone ⓘ https://orcid.org/0000-0002-9836-822X
Amy J Bastian ⓘ https://orcid.org/0000-0001-6079-0997

### Ethics

This study was approved by the Johns Hopkins School of Medicine Institutional Review Board and all participants, or their parent/legal guardian, provided informed consent prior to participation.

Reviewer #1 (Public review): https://doi.org/10.7554/eLife.101036.3.sa1
Reviewer #2 (Public review): https://doi.org/10.7554/eLife.101036.3.sa2
Reviewer #3 (Public review): https://doi.org/10.7554/eLife.101036.3.sa3
Author response https://doi.org/10.7554/eLife.101036.3.sa4

## Additional files

### Supplementary files

MDAR checklist

### Data availability

Behavioral data and scripts used for analysis and modeling are available on Mendeley Data.

The following dataset was generated:

| Author(s) | Year | Dataset title | Dataset URL | Database and Identifier |
|---|---|---|---|---|
| Hill N, Tripp H, Wolpert D, Malone L, Bastian A | 2025 | Developmental Reinforcement Motor Learning Dataset | http://doi.org/10.17632/g9z67hkyz5.1 | Mendeley Data, 10.17632/g9z67hkyz5.1 |

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

# Appendix 1

## Optimal exploration variability as a function of motor noise

Simulations (*Figure 7—figure supplement 2*) showed that the optimal exploration variability increases as a function of decreasing motor noise (as also observed with age in the data). Here we provide an intuition and proof for this inverse relation in a simplified setting. The setting is one in which there is a discrete point target offset at $+d$ from the desired reach (which without loss of generality we take to be at zero, $x_t = 0$; *Appendix 1—figure 1* left panel, red line).

As in the preferred model, we have two independent zero-mean Gaussian noise sources for exploration variability and motor noise:

$$e \sim \mathcal{N}(0, \sigma_e^2), \quad m \sim \mathcal{N}(0, \sigma_m^2).$$

We will examine how the expected reward depends on motor and exploration variability for the possible scenarios underlying a reach, that is (i) a reach after an unsuccessful trial and (ii) a reach after a successful trial that had exploration (i.e. the desired reach is updated). There is a third possibility, which is a reach after a successful trial that had no exploration (i.e. the previous trial was successful). However, the reaches in such a scenario do not involve exploration variability (only motor noise) and are not germane to the interplay of motor and exploration variability.

### Reach after an unsuccessful trial

After an unsuccessful reach, there is no updating of desired reach, and the next reach includes both exploration variability and motor noise, $(s_t = e_t + m_t)$. Therefore, the distribution of reach locations is given by $\mathcal{N}(0, \sigma_e^2 + \sigma_m^2)$ (Fig. 8 left panel, blue line). The probability of hitting the target (success, $r_t = 1$) is given by

$$P(r_t = 1 | s_t) = P(e_t + m_t = d)$$

In general, to maximize the density at $d$ away from the mean of a Gaussian distribution (*Appendix 1—figure 1* left panel, green line shows target location), the standard deviation should be $d$. Therefore, to maximize reward $d^2 = \sigma_e^2 + \sigma_m^2$. Rearranging gives $\sigma_e^2 = \max(0, d^2 - \sigma_m^2)$ and hence an inverse relation between optimal exploration variability and motor noise, provided $|d| > \sigma_m$ (if $|d| \leq \sigma_m$ it is better to have no exploration). This is the fundamental reason that exploration variability should decrease as motor noise increases, so as to maximize reward.

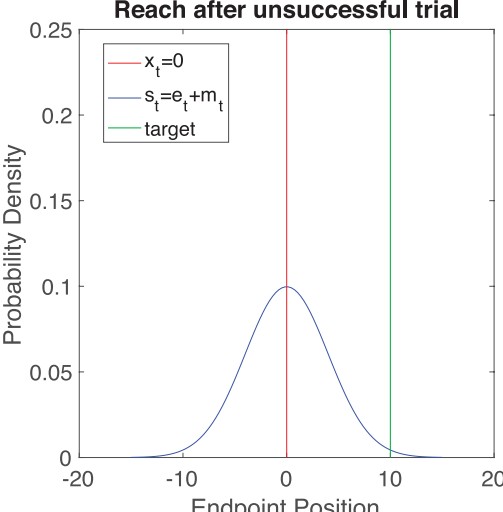 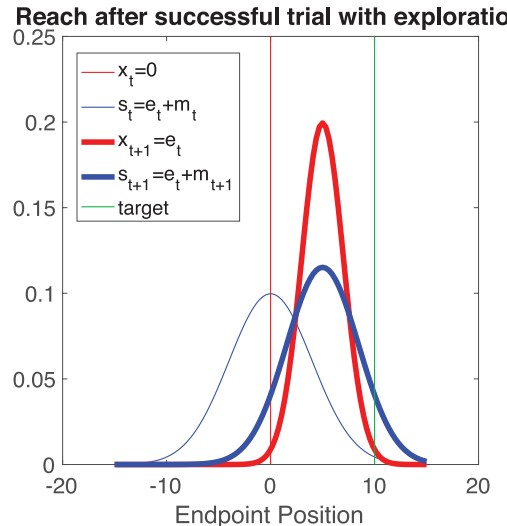

**Appendix 1—figure 1.** Distribution of desired reach and actual reach for a reach after an unsuccessful trial (left) and after a successful trial (right). Plots show probability density across endpoint positions. In both cases, the desired reach on trial $t$ is at $x_t = 0$ (thin red line shows distribution as a delta function). After an unsuccessful reach (left panel), the actual reach (blue distribution) includes exploration variability and motor noise. The probability of reward depends on the height of this distribution at the target ($d = 10$, shown by green line). For a successful trial with exploration (right panel), the distribution ($x_{t+1}$ and $s_t$, thin lines) is the same as for the left panel. However, the

desired reach is updated by exploration that led to success, which gives the distribution of the next desired reach ($x_{t+1}$, thick red line) and the next actual reach is this distribution with the addition of motor noise (thick blue line). For this illustration, both motor and exploration standard deviations were set to 8 and the target was set to +10 (green lines).

## Reach after a successful trial

The situation is more complicated if we consider what happens on the reach after a successful trial which had exploration. We need to consider (a) the update from the successful trial and (b) the probability of reward for the next trial.

(a) If the reach $s_t = e_t + m_t$ (*Appendix 1—figure 1* right panel, blue thin line) led to success ($e_t + m_t = d$) then the desired reach is updated by this exploration variability ($x_{t+1} = e_t$). We can consider the conditional distribution of $e_t$ given it led to success, which is known (from the simple properties of Gaussians) to be:

$$(e_t|m_t + e_t = d) = \mathcal{N}\left(\frac{\sigma_e^2}{\sigma_e^2 + \sigma_m^2}d, \frac{\sigma_m^2 \sigma_e^2}{\sigma_e^2 + \sigma_m^2}\right) \tag{15}$$

which is shown in *Appendix 1—figure 1* (right panel, red thick line). *Equation 15* shows that as $\sigma_e$ increases, so does the average update.

(b) The subsequent reach is only corrupted by motor noise (as the previous reach was successful) giving $s_{t+1} = e_t + m_{t+1}$. The reach distribution (*Appendix 1—figure 1* right panel, blue thick line) is therefore the same as *Equation 15* with the addition of motor noise:

$$\mathcal{N}\left(\frac{\sigma_e^2}{\sigma_e^2 + \sigma_m^2}d, \frac{\sigma_m^2 \sigma_e^2}{\sigma_e^2 + \sigma_m^2} + \sigma_m^2\right) \tag{16}$$

We can find the exploration variability that maximizes the reward, that is, maximizes the density at $d$. To do this, we can differentiate the Gaussian distribution with mean and variance given in *Equation 16* with respect to $\sigma_e$ to find the maximum. With a little algebra, this leads to the optimal exploration variability given by:

$$\sigma_e^2 = \frac{\sqrt{16\,d^4 + \sigma_m^4} + 4\,d^2 - 3\,\sigma_m^2}{4} \tag{17}$$

For an optimal $\sigma_e$ to exist, this expression must be positive (or the optimal $\sigma_e$ is 0) and this can be shown to be true provided $|d| > \sigma_m/\sqrt{3}$. How the optimal $\sigma_e$ changes with $\sigma_m$ depends on the sign of the derivative of *Equation 17* with respect to motor noise. We need only consider the components of *Equation 17* that depend on $\sigma_m$, that is:

$$f(\sigma_m) = \sqrt{16\,d^4 + \sigma_m^4} - 3\,\sigma_m^2$$

The derivative needs to be less than zero for the inverse relation to be held (i.e. increasing $\sigma_m$ reduces the optimal $\sigma_e$). That is:

$$f'(\sigma_m) = \frac{2\,\sigma_m^3}{\sqrt{16\,d^4 + \sigma_m^4}} - 6\,\sigma_m < 0$$

$$\frac{\sigma_m^2}{\sqrt{16\,d^4 + \sigma_m^4}} < 3$$

$$\sigma_m^4 < 9(16\,d^4 + \sigma_m^4)$$

$$0 < 144d^4 + 8\sigma_m^4$$

which is true. Therefore, the inverse relation persists even if we only consider reaches after a successful trial.

When the distance to the target is small compared to motor noise, then exploration actually reduces reward. However, in general for a real experiment with multiple reaches and exploration variability that does not change during learning, the above carries over to the full simulations (*Figure 7—figure supplement 2*).

# Appendix 2

## Supplementary Information

**Appendix 2—table 1.** Statistical analysis of sex, handedness, device, and browser on behavior. Results from one-way ANOVAs of participant-specific factors on precision from experimental block one (baseline) and distance from target from experimental block two (learning) for each of the four tasks. For all tasks, participant-specific factors did not significantly affect behavior.

**ANOVA Results**

| Continuous Probabilistic | | | |
|---|---|---|---|
| *Baseline Precision* | Sex | $F_{1,142}=0.251$ | p=0.617 |
| | Hand | $F_{1,142}=0.213$ | p=0.645 |
| | Device | $F_{2,141}=1.066$ | p=0.347 |
| | Browser | $F_{3,140}=0.414$ | p=0.743 |
| *Distance from Target* | Sex | $F_{1,142}=0.740$ | p=0.391 |
| | Hand | $F_{1,142}=0.815$ | p=0.368 |
| | Device | $F_{2,141}=0.733$ | p=0.482 |
| | Browser | $F_{3,140}=0.313$ | p=0.816 |
| **Discrete Probabilistic** | | | |
| *Baseline Precision* | Sex | $F_{1,137}=0.431$ | p=0.513 |
| | Hand | $F_{1,137}=0.923$ | p=0.338 |
| | Device | $F_{2,136}=0.677$ | p=0.510 |
| | Browser | $F_{3,135}=0.192$ | p=0.901 |
| *Distance from Target* | Sex | $F_{1,137}=0.096$ | p=0.758 |
| | Hand | $F_{1,137}=0.645$ | p=0.423 |
| | Device | $F_{2,136}=1.216$ | p=0.300 |
| | Browser | $F_{3,135}=0.998$ | p=0.396 |
| **Continuous Deterministic** | | | |
| *Baseline Precision* | Sex | $F_{1,48}=1.772$ | p=0.189 |
| | Hand | $F_{1,48}=0.544$ | p=0.464 |
| | Device | $F_{2,47}=0.337$ | p=0.716 |
| | Browser | $F_{3,46}=0.162$ | p=0.922 |
| *Distance from Target* | Sex | $F_{1,48}=0.455$ | p=0.503 |
| | Hand | $F_{1,48}=0.770$ | p=0.385 |
| | Device | $F_{2,47}=1.482$ | p=0.238 |
| | Browser | $F_{3,46}=0.343$ | p=0.794 |
| **Discrete Deterministic** | | | |
| *Baseline Precision* | Sex | $F_{1,50}=0.076$ | p=0.783 |
| | Hand | $F_{1,50}=0.015$ | p=0.902 |
| | Device | $F_{2,49}=0.691$ | p=0.506 |
| | Browser | $F_{2,49}=0.154$ | p=0.857 |

*Appendix 2—table 1 Continued on next page*

*Appendix 2—table 1 Continued*

**ANOVA Results**

| | | | |
|---|---|---|---|
| *Distance from Target* | Sex | $F_{1,50}=2.002$ | p=0.163 |
| | Hand | $F_{1,50}=0.105$ | p=0.747 |
| | Device | $F_{2,49}=0.933$ | p=0.400 |
| | Browser | $F_{2,49}=0.318$ | p=0.729 |

