## [Editor Report · eLife Assessment]

This **important** study tests the development of motor reinforcement learning from toddlerhood to adulthood, using a large online sample. They show that learning improves with age in a task that, like real-life movement, involves a continuous range of response options and probabilistic rewards, and link this shift to reduced movement variability and more efficient feedback-based learning through behavioural modeling. Simplifying the task with discrete actions and deterministic outcomes boosted younger children's performance, suggesting early learning is limited by spatial and probabilistic processing. The evidence is **convincing**, although future work may investigate more naturalistic movement.

---

## [Referee Report · Reviewer #1 (Public review)]

Summary:

Here the authors address how reinforcement-based sensorimotor adaptation changes throughout development. To address this question, they collected many participants in ages that ranged from small children (3 years old) to adulthood (18+ years old). The authors used four experiments to manipulate whether binary and positive reinforcement was provided probabilistically (e.g., 30 or 50%) versus deterministically (e.g.,100%), and continuous (infinite possible locations) versus discrete (binned possible locations) when the probability of reinforcement varied along the span of a large redundant target. The authors found that both movement variability and the extent of adaptation changed with age.

Strengths:

The major strength of the paper is the number of participants collected (n = 385). The authors also answer their primary question, that reinforcement-based sensorimotor adaptation changes throughout development, which was shown by utilizing established experimental designs and computational modelling. They have compared an extensive number of potential models, finding the one that best fits the data while penalizing the number of free parameters.

---

## [Referee Report · Reviewer #2 (Public review)]

Summary:

In this study, Hill and colleagues use a novel reinforcement-based motor learning task ("RML"), asking how aspects of RML change over the course of development from toddler years through adolescence. Multiple versions of the RML task were used in different samples, which varied on two dimensions: whether the reward probability of a given hand movement direction was deterministic or probabilistic, and whether the solution space had continuous reach targets or discrete reach targets. Using analyses of both raw behavioral data and model fits, the authors report four main results: First, developmental improvements reflected 3 clear changes, including increases in exploration, an increase in the RL learning rate, and a reduction of intrinsic motor noise. Second, changes to the task that made it discrete and/or deterministic both rescued performance in the youngest age groups, suggesting that observed deficits could be linked to continuous/probabilistic learning settings. Overall, the results shed light on how RML changes throughout human development, and the modeling characterizes the specific learning deficits seen in the youngest ages.

Strengths:

(1) This impressive work addresses an understudied subfield of motor control/psychology - the developmental trajectory of motor learning. It is thus timely and will interest many researchers.

(2) The task, analysis, and modeling methods are very strong. The empirical findings are rather clear and compelling, and the analysis approaches are convincing. Thus, at the empirical level, this study has very few weaknesses.

(3) The large sample sizes and in-lab replications further reflect the laudable rigor of the study.

(4) The main and supplemental figures are clear and concise.

---

## [Referee Report · Reviewer #3 (Public review)]

Summary:

The study investigates the development of reinforcement learning across the lifespan with a large sample of participants recruited for an online game. It finds that children gradually develop their abilities to learn reward probability, possibly hindered by their immature spatial processing and probabilistic reasoning abilities. Motor noise and exploration after a failure all contribute to children's subpar performance.

Strengths:

Experimental manipulations of both the continuity of movement options and the probabilistic nature of the reward function enable the inference of what cognitive factors differ between age groups.

A large sample of participants is studied.

The model-based analysis provides further insights into the development of reinforcement learning ability.

Weaknesses:

The conclusion that immature spatial processing and probabilistic reasoning abilities limit reinforcement learning here still needs more direct evidence.

---

## [Author Response]

The following is the authors’ response to the original reviews

Overview of changes in the revision

We thank the reviewers for the very helpful comments and have extensively revised the paper. We provide point-by-point responses below and here briefly highlight the major changes:

(1) We expanded the discussion of the relevant literature in children and adults.

(2) We improved the contextualization of our experimental design within previous reinforcement studies in both cognitive and motor domains highlighting the interplay between the two.

(3) We reorganized the primary and supplementary results to better communicate the findings of the studies.

(4) The modeling has been significantly revised and extended. We now formally compare 31 noise-based models and one value-based model and this led to a different model from the original being the preferred model. This has to a large extent cleaned up the modeling results. The preferred model is a special case (with no exploration after success) of the model proposed in Therrien et al. (2018). We also provide examples of individual fits of the model, fit all four tasks and show group fits for all, examine fits vs. data for the clamp phases by age, provide measures of relative and absolute goodness of fit, and examine how the optimal level of exploration varies with motor noise.

**Reviewer #1 (Public review):**
Summary:Here the authors address how reinforcement-based sensorimotor adaptation changes throughout development. To address this question, they collected many participants in ages that ranged from small children (3 years old) to adulthood (1 8+ years old). The authors used four experiments to manipulate whether binary and positive reinforcement was provided probabilistically (e.g., 30 or 50%) versus deterministically (e.g., 100%), and continuous (infinite possible locations) versus discrete (binned possible locations) when the probability of reinforcement varied along the span of a large redundant target. The authors found that both movement variability and the extent of adaptation changed with age.

Thank you for reviewing our work. One note of clarification. This work focuses on reinforcementbased learning throughout development but does not evaluate sensorimotor adaptation. The four tasks presented in this work are completed with veridical trajectory feedback (no perturbation).

The goal is to understand how children at different ages adjust their movements in response to reward feedback but does not evaluate sensorimotor adaptation. We now explain this distinction on line 35.

Strengths:The major strength of the paper is the number of participants collected (n = 385). The authors also answer their primary question, that reinforcement-based sensorimotor adaptation changes throughout development, which was shown by utilizing established experimental designs and computational modelling.

Thank you.

Weaknesses:Potential concerns involve inconsistent findings with secondary analyses, current assumptions that impact both interpr tation and computational modelling, and a lack of clearly stated hypotheses.(1) Multiple regression and Mediation Analyses.The challenge with these secondary analyses is that:(a) The results are inconsistent between Experiments 1 and 2, and the analysis was not performed for Experiments 3 and 4,(b) The authors used a two-stage procedure of using multiple regression to determine what variables to use for the mediation analysis, and(c)The authors already have a trial-by-trial model that is arguably more insightful.Given this, some suggested changes are to:(a) Perform the mediation analysis with all the possible variables (i.e., not informed by multiple regression) to see if the results are consistent.(b) Move the regression/mediation analysis to Supplementary, since it is slightly distracting given current inconsistencies and that the trial-by-trial model is arguably more insightful.

Based on these comments, we have chosen to remove the multiple regression and mediation analyses. We agree that they were distracting and that the trial-by-trial model allows for differentiation of motor noise from exploration variability in the learning block.

(2) Variability for different phases and model assumptions:A nice feature of the experimental design is the use of success and failure clamps. These clamped phases, along with baseline, are useful because they can provide insights into the partitioning of motor and exploratory noise. Based on the assumptions of the model, the success clamp would only reflect variability due to motor noise (excludes variability due to exploratory noise and any variability due to updates in reach aim). Thus, it is reasonable to expect that the success clamps would have lower variability than the failure clamps (which it obviously does in Figure 6), and presumably baseline (which provides success and failure feedback, thus would contain motor noise and likely some exploratory noise).However, in Figure 6, one visually observes greater variability during the success clamp (where it is assumed variability only comes from motor noise) compared to baseline where variability would come from: (a) Motor noise.(b) Likely some exploratory noise since there were some failures.(c) Updates in reach aim.

Thanks for this comment. It made us realize that some of our terminology was unintentionally misleading. Reaching to discrete targets in the Baseline block was done to (a) determine if participants could move successfully to targets that are the same width as the 100% reward zone in the continuous targets and (b) determine if there are age dependent changes in movement precision. We now realize that the term Baseline Variability was misleading and should really be called Baseline Precision.

This is an important distinction that bears on this reviewer's comment. In clamp trials, participants move to continuous targets. In baseline, participants move to discrete targets presented at different locations. Clamp Variability cannot be directly compared to Baseline Precision because they are qualitatively different. Since the target changes on each baseline trial, we would not expect updating of desired reach (the target is the desired reach) and there is therefore no updating of reach based on success or failure. The SD we calculate over baseline trials is the endpoint variability of the reach locations relative to the target centers. In success clamp, there are no targets so the task is qualitatively different.

We have updated the text to clarify terminology, expand upon our operational definitions, and motivate the distinct role of the baseline block in our task paradigm (line 674).

Given the comment above, can the authors please:(a) Statistically compare movement variability between the baseline, success clamp, and failure clamp phases.

Given our explanation in the previous point we don't think that comparing baseline to the clamp makes sense as the trials are qualitatively different.

(b) The authors have examined how their model predicts variability during success clamps and failure clamps, but can they also please show predictions for baseline (similar to that of Cashaback et al., 2019; Supplementary B, which alternatively used a no feedback baseline)?

Again, we do not think it makes sense to predict the baseline which as we mention above has discrete targets compared to the continuous targets in the learning phase.

(c) Can the authors show whether participants updated their aim towards their last successful reach during the success clamp? This would be a particularly insightful analysis of model assumptions.

We have now compared 31 models (see full details in next response) which include the 7 models in Roth et al. (2023). Several of these model variants have updating even after success with so called planning noise. We also now fit the model to the data that includes the clamp phases (we can't easily fit to success clamp alone as there are only 10 trials). We find that the preferred model is one that does not include updating after success.

(d) Different sources of movement variability have been proposed in the literature, as have different related models. One possibility is that the nervous system has knowledge of 'planned (noise)' movement variability that is always present, irrespective of success (van Beers, R.J. (2009). Motor learning is optimally tuned to the properties of motor noise. Neuron, 63(3), 406-417). The authors have used slightly different variations of their model in the past. Roth et al (2023) directly Rill compared several different plausible models with various combinations of motor, planned, and exploratory noise (Roth A, 2023, "Reinforcement-based processes actively regulate motor exploration along redundant solution manifolds." Proceedings of the Royal Society B 290: 20231475: see Supplemental). Their best-fit model seems similar to the one the authors propose here, but the current paper has the added benefit of the success and failure clamps to tease the different potential models apart. In light of the results of (a), (b), and (c), the authors are encouraged to provide a paragraph on how their model relates to the various sources of movement variability and ther models proposed in the literature.

Thank you for this. We realized that the models presented in Roth et al. (2023) as well as in other papers, are all special cases of a more general model. Moreover, in total there are 30 possible variants of the full model so we have now fit all 31 models to our larger datasets and performed model selection (Results and Methods). All the models can be efficiently fit by Kalman smoother to the actual data (rather than to summary statistics which has sometimes been done). For model selection, we fit only the 100 learning trials and chose the preferred model based on BIC on the children's data (Figure 5—figure Supplement 1). After selecting the preferred model we then refit this model to all trials including the clamps so as to obtain the best parameter estimates.

The preferred model was the same whether we combined the continuous and discrete probabilistic data or just examin d each task separately either for only the children or for the children and adults combined. The preferred model is a pecial case (no exploration after success) of the one proposed in Therrien et al. (2018) and has exploration variability (after failure) and motor noise with full updating with exploration variability (if any) after success. This model differs from the model in the original submission which included a partial update of the desired reach after exploration this was considered the learning rate. The current model suggests a unity learning rate.

In addition, as suggested by another reviewer, we also fit a value-based model which we adapted from the model described in Giron et al. (2023). This model was not preferred.

We have added a paragraph to the Discussion highlighting different sources of variability and links to our model comparison.

(e) line 155. Why would the success clamp be composed of both motor and exploratory noise? Please clarify in the text

This sentence was written to refer to clamps in general and not just success clamps. However, in the revision this sentence seemed unnecessary so we have removed it.

(3) Hypotheses:The introduction did not have any hypotheses of development and reinforcement, despite the discussion above setting up potential hypotheses. Did the authors have any hypotheses related to why they might expect age to change motor noise, exploratory noise, and learning rates? If so, what would the experimental behaviour look like to confirm these hypotheses? Currently, the manuscript reads more as an exploratory study, which is certainly fine if true, it should just be explicitly stated in the introduction. Note: on line 144, this is a prediction, not a hypothesis. Line 225: this idea could be sharpened. I believe the authors are speaking to the idea of having more explicit knowledge of action-target pairings changing behaviour.

We have included our hypotheses and predictions at two points in the paper In the introduction we modified the text to:

"We hypothesized that children's reinforcement learning abilities would improve with age, and depend on the developmental trajectory of exploration variability, learning rate (how much people adjust their reach after success), and motor noise (here defined as all sources of noise associated with movement, including sensory noise, memory noise, and motor noise). We think that these factors depend on the developmental progression of neural circuits that contribute to reinforcement learning abilities (Raznahan et al., 2014; Nelson et al., 2000; Schultz, 1998)."

In results we modified the sentence to:

"We predicted that discrete targets could increase exploration by encouraging children to move to a different target after failure.”

**Reviewer #2 (Public review):**
Summary:In this study, Hill and colleagues use a novel reinforcement-based motor learning task ("RML"), asking how aspects of RML change over the course of development from toddler years through adolescence. Multiple versions of the RML task were used in different samples, which varied on two dimensions: whether the reward probability of a given hand movement direction was deterministic or probabilistic, and whether the solution space had continuous reach targets or discrete reach targets. Using analyses of both raw behavioral data and model fits, the authors report four main results: First, developmental improvements reflected 3 clear changes, including increases in exploration, an increase in the RL learning rate, and a reduction of intrinsic motor noise. Second, changes to the task that made it discrete and/or deterministic both rescued performance in the youngest age groups, suggesting that observed deficits could be linked to continuous/probabilistic learning settings. Overall, the results shed light on how RML changes throughout human development, and the modeling characterizes the specific learning deficits seen in the youngest ages.Strengths:(1) This impressive work addresses an understudied subfield of motor control/psychology - the developmental trajectory of motor learning. It is thus timely and will interest many researchers.(2) The task, analysis, and modeling methods are very strong. The empirical findings are rather clear and compelling, and the analysis approaches are convincing. Thus, at the empirical level, this study has very few weaknesses.(3) The large sample sizes and in-lab replications further reflect the laudable rigor of the study.(4) The main and supplemental figures are clear and concise.

Thank you.

Weaknesses:(1) Framing.One weakness of the current paper is the framing, namely w/r/t what can be considered "cognitive" versus "non-cognitive" ("procedural?") here. In the Intro, for example, it is stated that there are specific features of RML tasks that deviate from cognitive tasks. This is of course true in terms of having a continuous choice space and motor noise, but spatially correlated reward functions are not a unique feature of motor learning (see e.g. Giron et al., 2023, NHB). Given the result here that simplifying the spatial memory demands of the task greatly improved learning for the youngest cohort, it is hard to say whether the task is truly getting at a motor learning process or more generic cognitive capacities for spatial learning, working memory, and hypothesis testing. This is not a logical problem with the design, as spatial reasoning and working memory are intrinsically tied to motor learning. However, I think the framing of the study could be revised to focus in on what the authors truly think is motor about the task versus more general psychological mechanisms. Indeed, it may be the case that deficits in motor learning in young children are mostly about cognitive factors, which is still an interesting result!

Thank you for these comments on the framing of our study. We now clearly acknowledge that all motor tasks have cognitive components (new paragraph at line 65). We also explain why we think our tasks has features not present in typical cognitive tasks.

(2) Links to other scholarship.If I'm not mistaken a common observation in tudies of the development of reinforcement learning is a decrease in exploration over-development (e.g., Nussenbaum and Hartley, 2019; Giron et al., 2023; Schulz et al., 2019); this contrasts with the current results which instead show an increase. It would be nice to see a more direct discussion of previous findings showing decreases in exploration over development, and why the current study deviates from that. It could also be useful for the authors to bring in concepts of different types of exploration (e.g. "directed" vs "random"), in their interpretations and potentially in their modeling.

We recognize that our results differ from prior work. The optimal exploration pattern differs from task to task. We now discuss that exploration is not one size fits all, it's benefits vary depending upon the task. We have added the following paragraphs to the Discussion section:

"One major finding from this study is that exploration variability increases with age. Some other studies of development have shown that exploration can decrease with age indicating that adults explore less compared to children (Schulz et al., 2019; Meder et al., 2021; Giron et al., 2023). We believe the divergence between our work and these previous findings is largely due to the experimental design of our study and the role of motor noise. In the paradigm used initially by Schulz et al. (2019) and replicated in different age groups by Meder et al. (2021) and Giron et al. (2023), participants push buttons on a two-dimensional grid to reveal continuous-valued rewards that are spatially correlated. Participants are unaware that there is a maximum reward available and therefore children may continue to explore to reduce uncertainty if they have difficulty evaluating whether they have reached a maxima. In our task by contrast, participants are given binary reward and told that there is a region in which reaches will always be rewarded. Motor noise is an additional factor which plays a key role in our reaching task but minimal if any role in the discretized grid task. As we show in simulations of our task, as motor noise goes down (as it is known to do through development) the optimal amount of exploration goes up (see Figure 7—figure Supplement 2 and Appendix 1). Therefore, the behavior of our participants is rational in terms of R230 increasing exploration as motor noise decreases.

A key result in our study is that exploration in our task reflects sensitivity to failure. Older children make larger adjustments after failure compared to younger children to find the highly rewarded zone more quickly. Dhawale et al. (2017) discuss the different contexts in which a participant may explore versus exploit (i.e., stick at the same position). Exploration is beneficial when reward is low as this indicates that the current solution is no longer ideal, and the participant should search for a better solution. Konrad et al. (2025) have recently shown this behavior in a real-world throwing task where 6 to 12 year old children increased throwing variability after missed trials and minimized variability after successful trials. This has also been shown in a postural motor control task where participants were more variable after non-rewarded trials compared to rewarded trials (Van Mastrigt et al., 2020). In general, these studies suggest that the optimal amount of exploration is dependent on the specifics of the task."

(3) Modeling.First, I may have missed something, but it is unclear to me if the model is actually accounting for the gradient of rewards (e.g., if I get a probabilistic reward moving at 45°, but then don't get one at 40°, I should be more likely to try 50° next then 35°). I couldn't tell from the current equations if this was the case, or if exploration was essentially "unsigned," nor if the multiple-trials-back regression analysis would truly capture signed behavior. If the model is sensitive to the gradient, it would be nice if this was more clear in the Methods. If not, it would be interesting to have a model that does "function approximation" of the task space, and see if that improves the fit or explains developmental changes.

The model we use (similar to Roth et al. (2023) and Therrien et al. (2016, 2018)) does not model the gradient. Exploration is always zero-mean Gaussian. As suggested by the reviewer, we now also fit a value-based model (described starting at line 810) which we adapted from the model presented in Giron et al. (2023). We show that the exploration and noise-based model is preferred over the value-based model.

The multiple-trials-back regression was unsigned as the intent was to look at the magnitude and not the direction of the change in movement. We have decided to remove this analysis from the manuscript as it was a source of confusion and secondary analysis that did not add substantially to the findings of these studies.

Second, I am curious if the current modeling approach could incorporate a kind of "action hysteresis" (aka perseveration), such that regardless of previous outcomes, the same action is biased to be repeated (or, based on parameter settings, avoided).

In some sense, the learning rate in the model in the original submission is highly related to perseveration. For example if the learning rate is 0, then there is complete perseveration as you simply repeat the same desired movement. If the rate is 1, there is no perseveration and values in between reflect different amounts of perseveration. Therefore, it is not easy to separate learning rate from perseveration. Adding perseveration as another parameter would likely make it and the learning unidentifiable. However, we now compare 31 models and those that have a non-unity learning rate are not preferred suggesting there is little perseveration.

(4) Psychological mechanisms. There is a line of work that shows that when children and adults perform RL tasks they use a combination of working memory and trial-by-trial incremental learning processes (e.g., Master et al., 2020; Collins and Frank 2012). Thus, the observed increase in the learning rate over development could in theory reflect improvements in instrumental learning, working memory, or both. Could it be that older participants are better at remembering their recent movements in short-term memory (Hadjiosif et al., 2023; Hillman et al., 2024)?

We agree that cognitive processes, such as working memory or visuospatial processing, play a role in our task and describe cognitive elements of our task in the introduction (new paragraph at line 65). However, the sensorimotor model we fit to the data does a good job of explaining the variation across age, which suggests that that age-dependent cognitive processes probably play a smaller role.

**Reviewer #3 (Public review):**
Summary:The study investigates reinforcement learning across the lifespan with a large sample of participants recruited for an online game. It finds that children gradually develop their abilities to learn reward probability, possibly hindered by their immature spatial processing and probabilistic reasoning abilities. Motor noise, reinforcement learning rate, and exploration after a failure all contribute to children's subpar performance.Strengths:(1) The paradigm is novel because it requires continuous movement to indicate people's choices, as opposed to discrete actions in previous studies.(2) A large sample of participants were recruited.(3) The model-based analysis provides further insights into the development of reinforcement learning ability.

Thank you.

Weaknesses:(1) The adequacy of model-based analysis is questionable, given the current presentation and some inconsistency in the results.

Thank you for raising this concern. We have substantially revised the model from our first submission. We now compare 31 noise-based models and 1 value-based model and fit all of the tasks with the preferred model. We perform model selection using the two tasks with the largest datasets to identify the preferred model. From the preferred model, we found the parameter fits for each individual dataset and simulated the trial by trial behavior allowing comparison between all four tasks. We now show examples of individual fits as well as provide a measure of goodness of fit. The expansion of our modeling approach has resolved inconsistencies and sharpened the conclusions drawn from our model.

(2) The task should not be labeled as reinforcement motor learning, as it is not about learning a motor skill or adapting to sensorimotor perturbations. It is a classical reinforcement learning paradigm.

We now make it clear that our reinforcement learning task has both motor and cognitive demands, but does not fall entirely within one of these domains. We use the term motor learning because it captures the fact that participants maximize reward by making different movements, corrupted by motor noise, to unmarked locations on a continuous target zone. When we look at previous ublications, it is clear that our task is similar to those that also refer to this as reinforcement motor learning Cashaback et al. (2019) (reaching task using a robotic arm in adults), Van Mastrigt et al. (2020) (weight shifting task in adults), and Konrad et al. (2025) (real-world throwing task in children). All of these tasks involve trial-by-trial learning through reinforcement to make the movement that is most effective for a given situation. We feel it is important to link our work to these previous studies and prefer to preserve the terminology of reinforcement motor learning.

**Recommendations for the authors:**

**Reviewing Editor Comments:**

Thank you for this summary. Rather than repeat the extended text from the responses to the reviewers here, we point the Editor to the appropriate reviewer responses for each issue raised.

The reviewers and editors have rated the significance of the findings in your manuscript as "Valuable" and the strength of evidence as "Solid" (see eLife evalutation). A consultancy discussion session to integrate the public reviews and recommendations per reviewer (listed below), has resulted in key recommendations for increasing the significance and strength of evidence:To increase the Significance of the findings, please consider the following:(1) Address and reframe the paper around whether the task is truly getting at a motor learning process or more generic cognitive decision-making capacities such as spatial memory, reward processing, and hypothesis testing.

We have revised the paper to address the comments on the framing of our work. Please see responses to the public review comments of Reviewers #2 and #3.

(2) It would be beneficial to specify the differences between traditional reinforcement algorithms (i.e., using softmax functions to explore, and build representations of state-action-reward) and the reinforcement learning models used here (i.e., explore with movement variability, update reach aim towards the last successful action), and compare present findings to previous cognitive reinforcement learning studies in children.

Please see response to the public review comments of Reviewer #1 in which we explain the expansion of our modeling approach to fit a value-based model as well as 31 other noise-based models. In our response to the public review comments of Reviewer #2, we comment on our expanded discussion of how our findings compare with previous cognitive reinforcement learning studies.

To move the "Strength of Evidence" to "Convincing", please consider doing the following:(1) Address some apparently inconsistent and unrealistic values of motor noise, exploration noise, and learning rate shown for individual participants e.g., Figure 5b; see comments reviewers 1 and take the following additional steps: plotting r squares for individual participants, discussing whether individual values of the fitted parameters are plausible and whether model parameters in each age group can extrapolate to the two clamp conditions and baselines.

We have substantially updated our modeling approach. Now that we compare 31 noise-based models, the preferred model does not show any inconsistent or unrealistic values (see response to Reviewer #3). Additionally, we now show example individual fits and provide both relative and absolute goodness of fit (see response to Reviewer #3).

(2) Relatedly, to further justify if model assumptions are met, it would be valuable to show that the current learning model fits the data better than alternative models presented in the literature by the authors themselves and by others (reviewer 1). This could include alternative development models that formalise the proposed explanations for age-related change: poor spatial memory, reward/outcome processing, and exploration strategies (reviewer 2).

Please see response to public review comments of Reviewer #1 in which we explain that we have now fit a value-based model as well as 31 other noise-based models providing a comparison of previous models as well as novel models. This led to a slightly different model being preferred over the model in the original submission (updated model has a learning rate of unity). These models span many of the processes previously proposed for such tasks. We feel that 32 models span a reasonable amount of space and do not believe we have the power to include memory issues or heuristic exploration strategies in the model.

(3) Perform the mediation analysis with all the possible variables (i.e., not informed by multiple regression) to see if the results are more consistent across studies and with the current approach (see comments reviewer 1).

Please see response to public review comments of Reviewer #1. We chose to focus only on the model based analysis because it allowed us to distinguish between exploration variability and motor noise.

Please see below for further specific recommendations from each reviewer.

**Reviewer #1 (Recommendations for the author):**
(1) In general, there should be more discussion and contextualization of other binary reinforcement tasks used in the motor literature. For example, work from Jeroen Smeets, Katinka van der Kooij, and Joseph Galea.

Thank you for this comment. We have edited the Introduction to better contextualize our work within the reinforcement motor learning literature (see line 67 and line 83).

(2) Line 32. Very minor. This sentence is fine, but perhaps could be slightly improved. “select a location along a continuous and infinite set of possible options (anywhere along the span of the bridge)"

Thank you for this comment. We have edited the sentence to reflect this suggestion.

(3) Line 57. To avoid some confusion in successive sentences: Perhaps, "Both children over 12 and adolescents...".

Thank you for this comment. We have edited the sentence to reflect this suggestion.

(4) Line 80. This is arguably not a mechanistic model, since it is likely not capturing the reward/reinforcement machinery used by the nervous system, such as updating the expected value using reward predic tion errors/dopamine. That said, this phenomenological model, and other similar models in the field, do very well to capture behaviour with a very simple set of explore and update rules.

We use mechanistic in the standard use in modeling, as in Levenstein et al. (2023), for example. The contrast is not with neural modeling, but with normative modeling, in which one develops a model to optimize a function (or descriptive models as to what a system is trying to achieve). In mechanistic modeling one proposes a mechanism and this can be at a state-space level (as in our case) or a neural level (as suggested my the reviewer) but both are considered mechanistic, just at different levels. Quoting Levenstein "... mechanistic models, in which complex processes are summarized in schematic or conceptual structures that represent general properties of components and their interactions, are also commonly used." We now reference the Levenstein paper to clarify what we mean by mechanistic.

(5) Figure 1. It would be useful to state that the x-axis in Figure 1 is in normalized units, depending on the device.

Thank you for this comment. We have added a description of the x-axis units to the Fig. 1 caption.

(6) Were there differences in behaviour for these different devices? e.g., how different was motor noise for the mouse, trackpad, and touchscreen?

Thank you for this question. We did not find a significant effect of device on learning or precision in the baseline block. We have added these one way ANOVA results for each task in Supplementary Table 1.

(7) Line 98. Please state that participants received reinforcement feedback during baseline.

Thank you for this comment. We have updated the text to specify that participants receive reward feedback during the baseline block.

(8) Line 99. Did the distance from the last baseline trial influence whether the participant learned or did not learn? For example, would it place them too far from the peak success location such that it impacted learning?

Thank you for this question. We looked at whether the position of movement on the last baseline block trial was correlated with the first movement position in the learning block. We did not find any correlations between these positions for any of the tasks. Interestingly, we found that the majority of participants move to the center of the workspace on the first trial of the learning block for all tasks (either in the presence of the novel continuous target scene or the presentation of 7 targets all at once). We do not think that the last movement in the baseline block "primed" the participant for the location of the success zone in the learning block. We have added the following sentence to the Results section:

"Note that the reach location for the first learning trial was not affected by (correlated with) the target position on the last baseline trial (p > 0.3 for both children and adults, separately)."

(9) The term learning distance could be improved. Perhaps use distance from target.

Thank you for this comment. We appreciate that learning distance defined with 0 as the best value is counter intuitive. We have changed the language to be "distance from target" as the learning metric.

(10) Line 188. This equation is correct, but to estimate what the standard deviation by the distribution of changes in reach position is more involved. Not sure if the authors carried out this full procedure, which is described in Cashaback et al., 2019; Supplemental 2.

There appear to be no Supplemental 2 in the referenced paper so we assume the reviewer is referring to Supplemental B which deals with a shuffling procedure to examine lag-1 correlations.

In our tasks, we are limited to only 9 trials to analyze in each clamp phase so do not feel a shuffling analysis is warranted. In these blocks, we are not trying to 'estimate what the standard deviation by the distribution of changes in reach position' but instead are calculating the standard deviation of the reach locations and comparing the model fit (for which the reviewer says the formula is correct) with the data. We are unclear what additional steps the reviewer is suggesting. In our updated model analysis, we fit the data including the clamp phases for better parameter estimation. We use simulations to estimate s.d. in the clamp phase (as we ensure in simulations the data does not fall outside the workspace) making the previous analytic formulas an approximation that are no longer used.

(11) Line 197-199. Having done the demo task, it is somewhat surprising that a 3-year-old could understand these instructions (whose comprehension can be very different from even a 5-year old).

Thank you for raising this concern. We recognize that the younger participants likely have different comprehension levels compared to older participants. However, we believe that the majority of even the youngest participants were able to sufficiently understand the goal of the task to move in a way to get the video clip to play. We intentionally designed the tasks to be simple such that the only instructions the child needed to understand were that the goal was to get the video clip to play as much as possible and the video clip played based on their movement. Though the majority of younger children struggled to learn well on the probabilistic tasks, they were able to learn well on the deterministic tasks where the task instructions were virtually identical with the exception of how many places in the workspace could gain reward. On the continuous probabilistic task, we did have a small number (n = 3) of 3 to 5 year olds who exhibited more mature learning ability which gives us confidence that the younger children were able to understand the task goal.

(12) Line 497: Can the authors please report the F-score and p-value separately for each of these one-way ANOVA (the device is of particular interest here).

Thank you for this request. We have added ina upplementarytable (Supplementary Table 1) with the results of these ANOVAs.

(13) Past work has discussed how motivation influences learning, which is a function of success rate (van der Kooij, K., in 't Veld, L., & Hennink, T. (2021). Motivation as a function of success frequency. Motivation and Emotion, 45, 759-768.). Can the authors please discuss how that may change throughout development?

Thank you for this comment. While motivation most probably plays a role in learning, in particular in a game environment, this was out of the scope of the direct focus of this work and not something that our studies were designed to test. We have added the following sentence to the discussion section to address this comment:

"We also recognize that other processes, such as memory and motivation, could affect performance on these tasks however our study was not designed to test these processes directly and future work would benefit from exploring these other components more explicitly."

(14) Supplement 6. This analysis is somewhat incomplete because it does not consider success.Pekny and collegues (2015) looked at 3 trials back but considered both success and reward. However, their analysis has issues since successive time points are not i.i.d., and spurious relationships can arise. This issue is brought up by Dwahale (Dhawale, A. K., Miyamoto, Y. R., Smith, M. A., & R475 Ölveczky, B. P. (2019). Adaptive regulation of motor variability. Current Biology, 29(21), 3551-3562.). Perhaps it is best to remove this analysis from the paper.

Thank you for this comment. We have decided to remove this secondary analysis from the paper as it was a source of confusion and did not add to the understanding and interpretation of our behavioral results.

**Reviewer #2 (Recommendations for the author):**
(1) the path length ratio analyses in the supplemental are interesting but are not mentioned in the main paper. I think it would be helpful to mention these as they are somewhat dramatic effects

Thank you for this comment. Path length ratios are defined in the Methods and results are briefly summarized in the Results section with a point to the supplementary figures. We have updated the text to more explicitly report the age related differences in path length ratios.

(2) The second to last paragraph of the intro could use a sentence motivating the use ofthe different task features (deterministic/probabilistic and discrete/continuous).

Thank you for this comment. We have added an additional motivating sentence to the introduction.

**Reviewer #3 (Recommendations for the author):**
The paper labeled the task as one for reinforcement motor learning, which is not quite appropriate in my opinion. Motor learning typically refers to either skill learning or motor adaptation, the former for improving speed-accuracy tradeoffs in a certain (often new) motor skill task and the latter for accommodating some sensorimotor perturbations for an existing motor skill task. The gaming task here is for neither. It is more like a

decision-making task with a slight contribution to motor execution, i.e., motor noise. I would recommend the authors label the learning as reinforcement learning instead of reinforcement motor learning.

Thank you for this comment. As noted in the response to the public review comments, we agree that this task has components of classical reinforcement learning (i.e. responding to a binary reward) but we specifically designed it to require the learning of a movement within a novel game environment. We have added a new paragraph to the introduction where we acknowledge the interplay between cognitive and motor mechanisms while also underscoring the features in our task that we think are not present in typical cognitive tasks.

My major concern is whether the model adequately captures subjects' behavior and whether we can conclude with confidence from model fitting. Motor noise, exploration noise, and learning rate, which fit individual learning patterns (Figure 5b), show some quite unrealistic values. For example, some subjects have nearly zero motor noise and a 100% learning rate.

We have now compared 31 models and the preferred model is different from the one in the first submission. The parameter fits of the new model do not saturate in any way and appear reasonable to us. The updates to the model analysis have addressed the concern of previously seen unrealistic values in the prior draft.

Currently, the paper does not report the fitting quality for individual subjects. It is good to have an exemplary subject's fit shown, too. My guess is that the r-squared would be quite low for this type of data. Still, given that the children's data is noisier, it might be good to use the adult data to show how good the fitting can be (individual fits, r squares, whether the fitted parameters make sense, whether it can extrapolate to the two clamp phases). Indeed, the reliability of model fitting affects how we should view the age effect of these model parameters.

We now show fits to individual subjects. But since this is a Kalman smoother it fits the data perfectly by generating its best estimate of motor noise and exploration variability on each trial to fully account for the data — so in that sense *R*^2^ is always 1 so that is not helpful.

While the BIC analysis with the other model variants provides a relative goodness of fit, it is not straightforward to provide an absolute goodness of fit such as standard *R*^2^ for a feedforward simulation of the model given the parameters (rather than the output of the Kalman smoother). There are two problems. First, there is no single model output. Each time the model is simulated with the fit parameters it produces a different output (due to motor noise, exploration variability and reward stochasticity). Second, the model is not meant to reproduce the actual motor noise, exploration variability and reward stochasticity of a trial. For example, the model could fit pure Gaussian motor noise across trials (for a poor learner) by accurately fitting the standard deviation of motor noise but would not be expected to actually match each data point so would have a traditional *R*^2^ of O.

To provide an overall goodness of fit we have to reduce the noise component and to do so we exam ined the traditional *R*^2^ between the average of all the children's data and the average simulation of the model (from the median of 1000 simulations per participant) so as to reduce the stochastic variation. The results for the continuous probabilistic and discrete probabilistic task are *R*^2^ of 0.41 and 0.72, respectively.

Not that variability in the "success clamp" doe not change across ages (Figure 4C) and does not contribute to the learning effect (Figure 4F). However, it is regarded as reflecting motor noise (Figure SC), which then decreases over age from the model fitting (Figure 5B). How do we reconcile these contradictions? Again, this calls the model fitting into question.

For the success clamp, we only have 9 trials to calculate variability which limits our power to detect significance with age. In contrast, the model uses all 120 trials to estimate motor noise. There is a downward trend with age in the behavioral data which we now show overlaid on the fits of the model for both probabilistic conditions (Figure 5—figure Supplement 4) and Figure 6—figure Supplement 4. These show a reasonable match and although the variance explained is 1 6 and 56% (we limit to 9 trials so as to match the fail clamp), the correlations are 0.52 and 0.78 suggesting we have reasonable relation although there may be other small sources of variability not captured in the model.

Figure 5C: it appears one bivariate outlier contributes a lot to the overall significant correlation here for the "success clamp".

Recalculating after removing that point in original Fig 5C was still significant and we feel the plots mentioned in the previous point add useful information to this issue. With the new model this figure has changed.

It is still a concern that the young children did not understand the instructions. Nine 3-to-8 children (out of 48) were better explained by the noisy only model than the full model. In contrast, ten of the rest of the participants (out of 98) were better explained by the noisy-only model. It appears that there is a higher percentage of the "young" children who didn't get the instruction than the older ones.

Thank you for this comment. We did take participant comprehension of the task into consideration during the task design. We specifically designed it so that the instructions were simple and straight forward. The child simply needs to understand the underlying goal to make the video clip play as often as possible and that they must move the penguin to certain positions to get it to play. By having a very simple task goal, we are able to test a naturalistic response to reinforcement in the absence of an explicit strategy in a task suited even for young children.

We used the updated reinforcement learning model to assess whether an individual's performance is consistent with understanding the task. In the case of a child who does not understand the task, we expect that they simply have motor noise on their reach, and crucially, that they would not explore more after failure, nor update their reach after success. Therefore, we used a likelihood ratio test to examine whether the preferred model was significantly better at explaining each participant's data compared to the model variant which had only motor noise (Model 1). Focusing on only the youngest children (age 3-5), this analysis showed that that 43, 59, 65 and 86% of children (out of N = 21, 22, 20 and 21) for the continuous probabilistic, discrete probabilistic, continuous deterministic, and discrete deterministic conditions, respectively, were better fit with the preferred model, indicating non-zero exploration after failure. In the 3-5 year old group for the discrete deterministic condition, 18 out of 21 had performance better fit by the preferred model, suggesting this age group understands the basic task of moving in different directions to find a rewarding location.

The reduced numbers fit by the preferred model for the other conditions likely reflects differences in the task conditions (continuous and/or probabilistic) rather than a lack of understanding of the goal of the task. We include this analysis as a new subsection at the end of the Results.

Supplementary Figure 2: the first panel should belong to a 3-year-old not a 5-year-old? How are these panels organized? This is kind of confusing.

Thank you for this comment. Figure 2—figure Supplement 1 and Figure 2—figure Supplement 2 are arranged with devices in the columns and a sample from each age bin in the rows. For example in Figure 2—figure Supplement 1, column 1, row 1 is a mouse using participant age 3 to 5 years old while column 3, row 2 is a touch screen using participant age 6 to 8 years old. We have edited the labeling on both figures to make the arrangement of the data more clear.

Line 222: make this a complete sentence.

This sentence has been edited to a complete sentence.

Line 331: grammar.

This sentence has been edited for grammar.